# The occurrence of 'Sleeping Beauty' publications in medical research: Their scientific impact and technological relevance

**Anthony F. J. van Raan** [ID]*, **Jos J. Winnink**

Centre for Science and Technology Studies, Leiden University, Leiden, The Netherlands

* vanraan@cwts.leidenuniv.nl

**Data Availability Statement:** All relevant data are within the manuscript and its Supporting Information files. In addition files with basic data

## Abstract

We investigate publications in medical research that have gone unnoticed for a number of years after being published and then suddenly become cited to a significant degree. Such publications are called Sleeping Beauties (SBs). This study focuses on SBs that are cited in patents. We find that the increasing trend of the relative number of SBs comes to an end around 1998. However, still a constant fraction of publications becomes an SB. Many SBs become highly cited publications, they even belong to the top-10 to 20% most cited publications in their field. We measured the scaling of the number of SBs in relation to the sleeping period length, during-sleep citation-intensity, and with awakening citation-intensity. We determined the Grand Sleeping Beauty Equation for these medical SBs which shows that the probability of awakening after a period of deep sleep is becoming rapidly smaller for longer sleeping periods and that the probability for higher awakening intensities decreases extremely rapidly. The exponents of the scaling functions show a time-dependent behavior which suggests a decreasing occurrence of SBs with longer sleeping periods. We demonstrate that the fraction of SBs cited by patents before scientific awakening exponentially increases. This finding shows that the technological time lag is becoming shorter than the sleeping time. Inventor-author self-citations may result in shorter technological time lags, but this effect is small. Finally, we discuss characteristics of an SBs that became one of the highest cited medical papers ever.

## Introduction

In science, a Sleeping Beauty is a publication that goes largely or completely unnoticed ('sleeps') for a long time and then, almost suddenly, attracts a lot of attention ('is awakened by a prince'). Garfield focused the attention on the phenomenon of 'delayed recognition' [1, 2, 3, 4] which was linked to 'premature discovery' or 'being ahead of time' [5]. We refer to our earlier work [6] for an overview of the literature on Sleeping Beauties (SBs) and for an extensive analysis of Sleeping Beauties in physics, chemistry, and engineering & computer science. More than half of the SBs appeared to be application-oriented.

will be uploaded in the Open Science Framework, osf.io/4ru96.

**Funding:** The authors received no specific funding for this work.

**Competing interests:** The authors have declared that no competing interests exist.

In follow-up work [7, 8], we investigated whether the Sleeping Beauties in physics, chemistry, and engineering & computer science are also cited in patents, i.e., SBs that appear as scientific non-patent references (SNPR) in patents [9]. We found that the time lag between the publication year of the SB-SNPRs and their first citation in a patent was becoming shorter in recent years. In this paper we investigate this phenomenon for medical research. To find out how this phenomenon differs between natural science research and medical research.

To make a comparison of the results in this study for the medical research fields with our previous results [6, 7, 8] for the natural sciences possible, we follow similar procedures for data collection and data analysis. We used the WOS categories presented in S1 Table to delineate the medical research fields. After discussing the identification of medical SBs on the basis of a set of measurement variables and the calculation of numbers and trends as a function of time, we analyze the scaling of the number of SBs with sleeping period, sleeping intensity and awakening citation intensity. We then identify those SBs that are cited in patents (SB-SNPRs) and analyze the time lag between publication year and year of the first patent citation. In this context we also focus on the role of inventor-author combinations. Finally, we discuss the characteristics of an extremely highly cited medical SB-SNPR in particular on the basis of co-citation and bibliographic coupling maps.

## Materials and methods

### Data sources

In this study we use the following data sources: (1) the CWTS in-house enhanced Web of Science (WoS) database, and (2) the Spring 2016 version of the PATSTAT database. The CWTS in-house version of the WOS is compatible with the on-line version of the WoS provided by Clarivate Analytics but is limited to publications from 1980 onwards. An advantage of the in-house version of the WoS is that it is possible to cross-link information from different databases. The PATSTAT database covers information of all relevant patents worldwide and is provided by the European Patent Office especially for statistical patent analysis.

### Measurement variables, choice of sets, identification of sleeping beauties

With a fast and efficient search algorithm written in SQL which can be applied to the CWTS enhanced Web of Science (WoS) database (starting year 1980) [6, 7, 8] we can tune four main variables: (1) *length of the sleep* in years after publication (sleeping period $s$); (2) *depth of sleep* in terms of the citation rate during the sleeping period ($c_s$); (3) *awakening* period in years after the sleeping period ($a$); and (4) *awakening citation-intensity* in terms of the citation rate during the awakening period ($c_a$). We define $c_s = 0$ as a coma, $c_s$ between 0.1 and 0.5 as a very deep sleep, and $c_s$ between 0.6 and 1.0 as a deep sleep. In the algorithm we can apply a threshold value $c_s(\max)$ for the citation rate during the sleeping period. For instance, if we take $c_s(\max) = 1.0$ we cover the range from $c_s = 0$ (coma) to $c_s = 1.0$ (deep sleep). In this study we use a five-years awakening period immediately after the sleeping period, i.e., $a(\max) = a(\min) = 5$. Furthermore, we require that the SBs must have an awakening intensity of at least, on average, 5 citations per year, i.e., $c_a(\min) = 5.0$. Thus, for a complete analysis of the SBs, we need a total measuring period equal to sleeping period plus awakening period of five years ($a(\max) = a(\min) = 5$). As a consequence, the longer $s$, the less publication years we have for our analysis. For instance, if $s = 20$, we need a time period 20+5 = 25 years, and given that 2016 is the last year of the citation measurements only SBs (with $s = 20$) published between 1980 and 1992 can be considered. On the other hand, for SBs with $s = 5$ we need 5+5 = 10 years and thus 2007 is in this case the last publication year of the measurement.

The above definition of sleeping and awakening period can be written as follows. Given that $t_1$ is the year of publication and $c(t_i)$ is the number of citations (excluding self-citations) in any year $t_i$ then if

$$\{c(t_1) + \ldots\ldots c(t_n)\}/n \leq 1.0 \; and \; \{c(t_{n+1}) + \ldots\ldots c(t_{n+5})\}/5 \geq 5.0$$

the sleeping time is n years in time period $[t_1, t_n]$ and the subsequent time period $[t_{n+1}, t_{n+5}]$ is the awakening period.

We identify all publications covered by the WoS in all medical research fields (see S1 Table) that meet the parameters in Table 1. In this way we find all SBs with different sleeping periods, citation rates during sleep up to 1.0 and citation rates from 5.0 during the awakening period of 5 years in the given range of publication years. On the basis of these data we determine the annual numbers of these SBs. In order to identify these SBs, the search algorithm had to calculate for about 7.2 million publications their complete citation history (1980–2017) covering 230 million citations.

In order to give a first overall impression of quantities, we also give in Table 1 the total number of identified SBs with a specific sleeping time within the relevant range of publication years. An interesting question is: how many publications are there with the *same* sleeping characteristics as the SBs but with awakening citation-intensities ($c_a$) *below* the threshold used for the SBs? Thus, we identified all publications in the period 1980–2007 with $c_s(\text{max}) = 1.0$ during the first 5 years after publication ($s = 5$) *and* $c_a(\text{max}) = 4.9$ during the sixth to tenth year after publication ($a(\text{max}) = a(\text{min}) = 5$). This number is 4,281,407. Thus, the probability that a publication with no or only a few citations in the first five years after publication will become a Sleeping Beauty is about 1 thousandth.

## Basic numbers and trends

In this section, we focus on the quantitative characteristics of SBs in more detail. The total number of WoS-covered medical research publications increases considerably over the entire measuring period. It is therefore plausible that also the number of SBs will increase. Clearly it is important to normalize the measured number of SBs: if the number of SBs increases less in comparison to the total number of publications, we have an indication that the probability for a publication to become a SB decreases.

We first counted the total annual number of all medical publications $N_{tot}$ for the period 1980–2008 (the CWTS in-house version of the WoS contains publications from 1980 onwards). The results are given in Table 2 and in Fig 1. We use 2000 as index year to calculate the growth factor. We find an exponential growth of the WoS-covered medical research literature of about 3% per year.

The annual numbers $N(s)$ of SBs are determined with our SQL search algorithm. These numbers are given in S2 Table for all SBs with $s = 5, 10, 15, 20, 25,$ and 30 and for five values of

**Table 1. Variables and numbers.**

| s | $c_s$ (max) | $a$(max) = $a$(min) | $c_a$ (min) | pub years | N |
|---|---|---|---|---|---|
| 5 | 1.0 | 5 | 5.0 | 1980–2007 | 5,247 |
| 10 | 1.0 | 5 | 5.0 | 1980–2002 | 614 |
| 15 | 1.0 | 5 | 5.0 | 1980–1997 | 199 |
| 20 | 1.0 | 5 | 5.0 | 1980–1992 | 110 |
| 25 | 1.0 | 5 | 5.0 | 1980–1987 | 62 |
| 30 | 1.0 | 5 | 5.0 | 1980–1982 | 19 |

**Table 2. Number of WoS-covered publications in the medical research fields.**

| publication year | $N_{tot}$ | growth factor | publication year | $N_{tot}$ | growth factor |
|---|---|---|---|---|---|
| 1980 | 149,200 | 0.51 | 1995 | 236,552 | 0.82 |
| 1981 | 155,728 | 0.54 | 1996 | 275,168 | 0.95 |
| 1982 | 164,595 | 0.57 | 1997 | 280,090 | 0.97 |
| 1983 | 171,317 | 0.59 | 1998 | 286,664 | 0.99 |
| 1984 | 176,770 | 0.61 | 1999 | 290,279 | 1.00 |
| 1985 | 184,079 | 0.63 | 2000 | 289,966 | 1.00 |
| 1986 | 191,139 | 0.66 | 2001 | 286,523 | 0.99 |
| 1987 | 195,131 | 0.67 | 2002 | 289,944 | 1.00 |
| 1988 | 201,620 | 0.70 | 2003 | 298,903 | 1.03 |
| 1989 | 209,850 | 0.72 | 2004 | 309,131 | 1.07 |
| 1990 | 214,170 | 0.74 | 2005 | 333,650 | 1.15 |
| 1991 | 220,457 | 0.76 | 2006 | 352,924 | 1.22 |
| 1992 | 224,072 | 0.77 | 2007 | 379,239 | 1.31 |
| 1993 | 224,123 | 0.77 | 2008 | 406,619 | 1.40 |
| 1994 | 224,721 | 0.77 | | | |

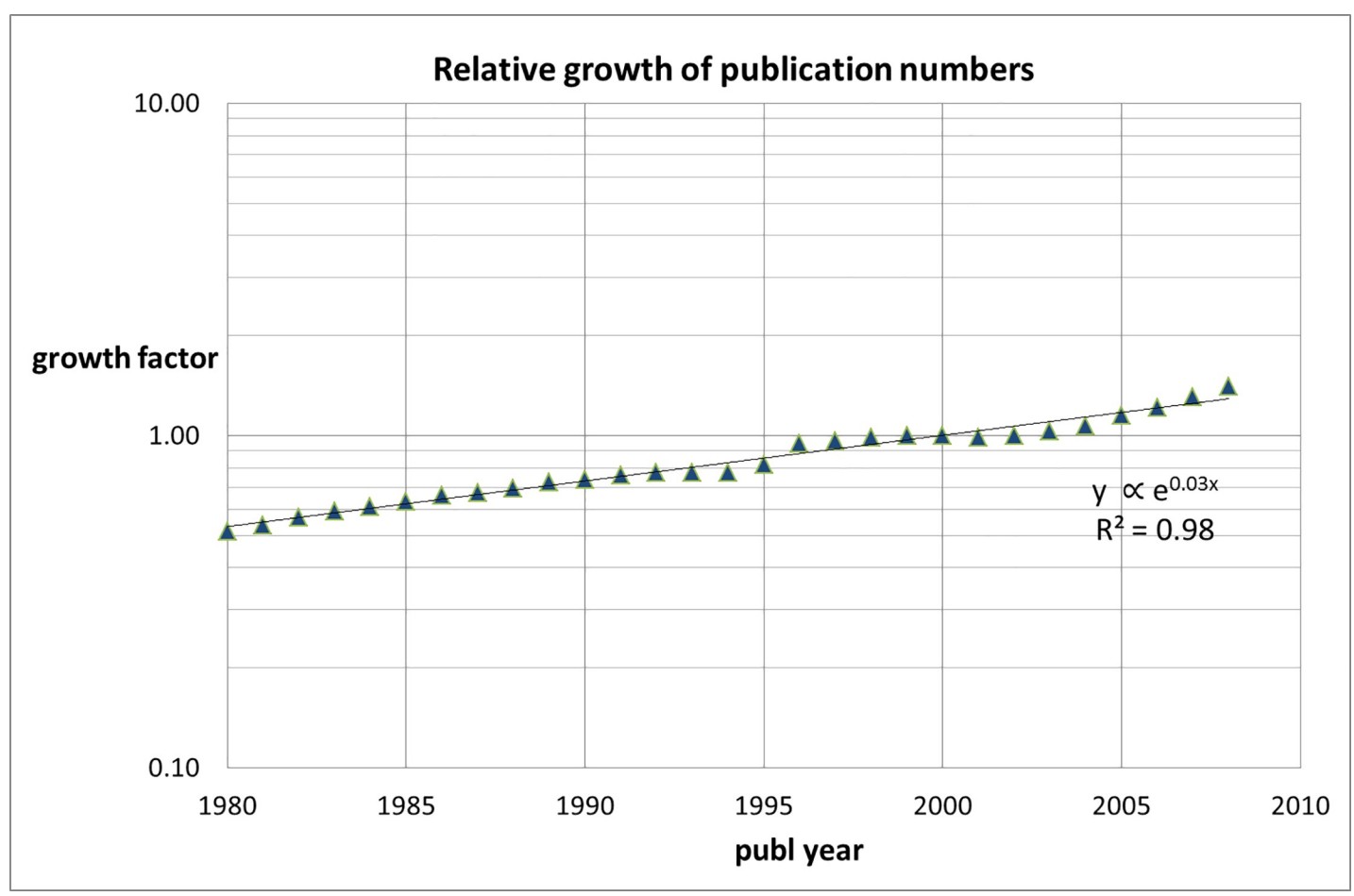

**Fig 1. Trend of the number of WoS-covered publications in the medical research fields (index: 2000).**

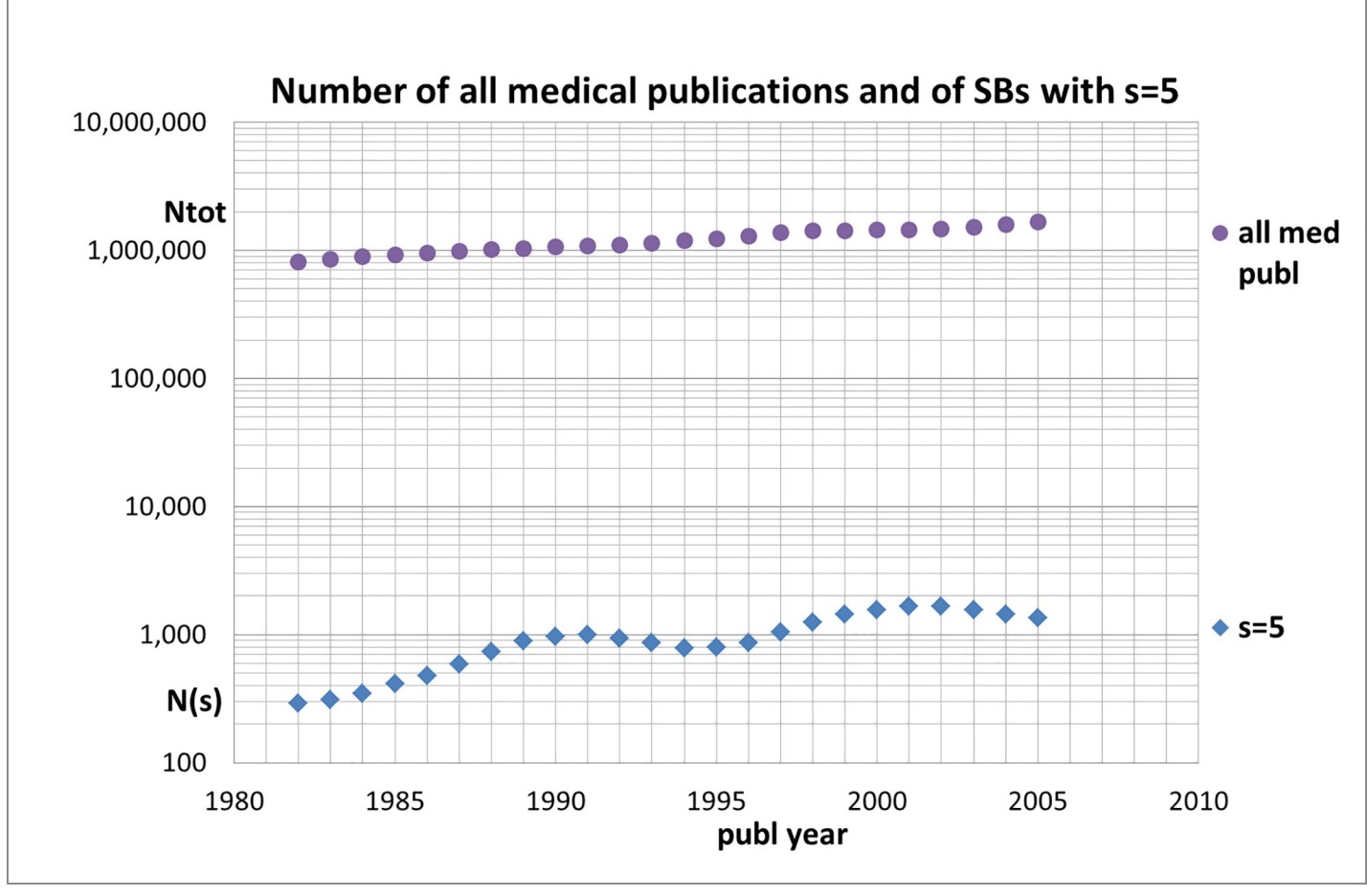

**Fig 2. Trend of the number (real, i.e. not-normalized) of all medical publications ($N_{tot}$), and the number N(s) of SBs with sleeping time $s$ = 5; numbers are the totals of successive five-years blocks, each five-years block is located in the figure by its middle year.**

$c_s$(max). These numbers are what they are but if we want to find out whether the number of SBs is increasing or not, we have to *normalize* the numbers relative to the total number of all medical papers, i.e., normalization on the basis of the growth factors (i.e., dividing N(s) by the growth factor of the relevant year), see Table 2. As an example, we show in S3 Table the results for SBs with $s$ = 5. The effect of normalization is shown in Figs 2 and 3. In Fig 2 the trend of the real (not-normalized) numbers of all medical publications and of the SBs with sleeping time $s$ = 5 is given. Notice that the number of SBs with $s$ = 5 is about three orders of magnitudes lower than the total number of medical publications. We see that from the late 1990s the real number of SBs shows a sort of oscillating behavior rather than a clear increase whereas the total number of medical papers covered by the WoS still increases. This effect is even more clear if we normalize the numbers as discussed above, see Fig 3 where we also included the normalized numbers n(s) for SBs with $s$ = 10 and 15.

The SBs with $s$ = 5 have the shortest sleeping period and thus they can be analyzed for the most recent times, until 2007. The numbers of these SBs show considerable fluctuations during the measuring period. Looking at the normalized numbers, we find that the relative occurrence of SBs with $s$ = 5 doubled since the early 1980s with an average increase of 4% up to about the late 1990s. But in the more recent years the increase of the relative occurrence of SBs with $s$ = 5 comes to a halt. In fact, the relative occurrence of SBs with $s$ = 5 in 2007 is about the same as

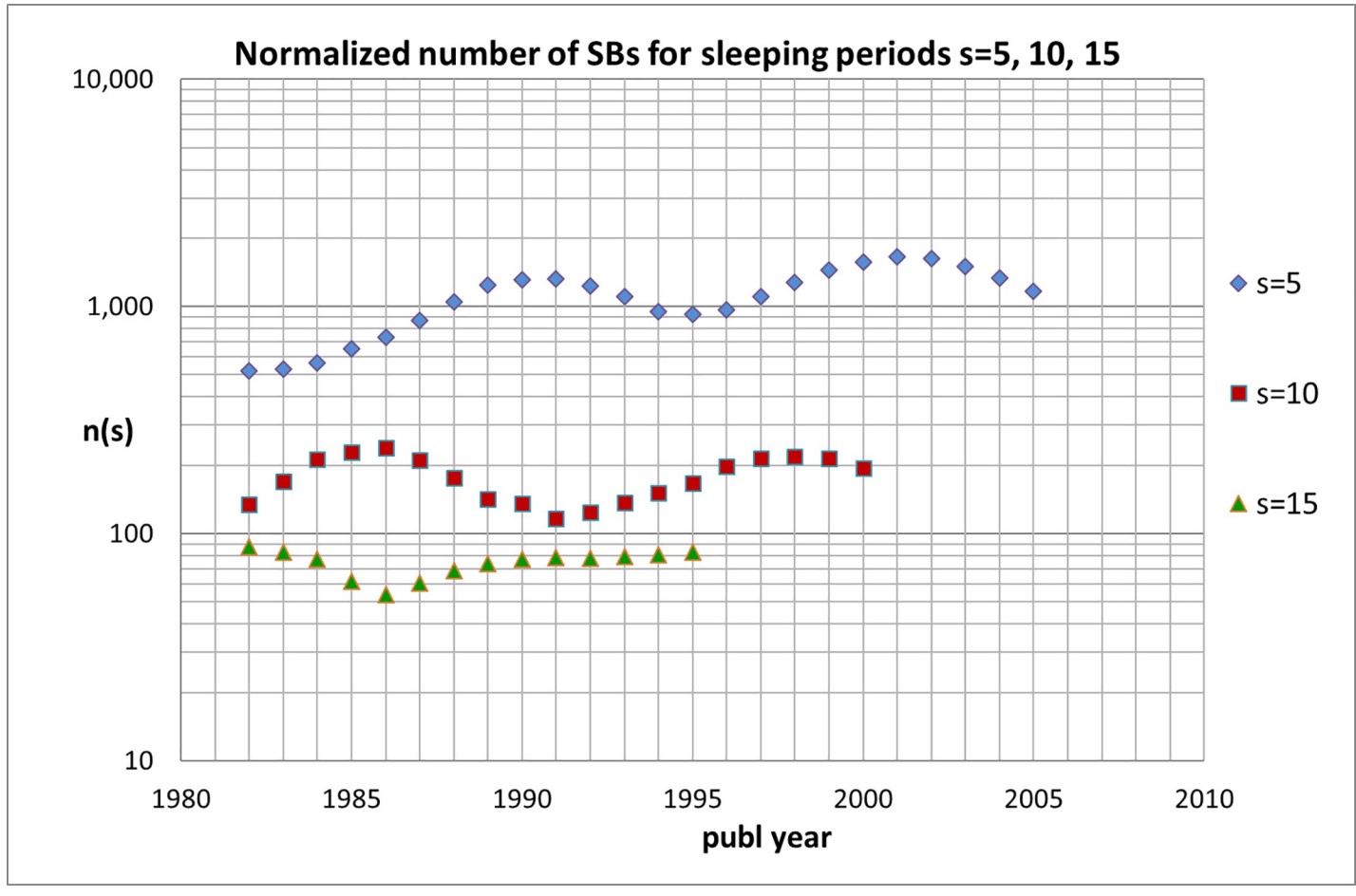

**Fig 3. Trend of the normalized numbers n(s) of SBs in the medical research fields; numbers are the totals of successive five-years blocks, each five-years block is located in the figure by its middle year.**

twenty years before. It confirms our observations in the natural science research fields and supports our conjecture that the expanding worldwide facilities to access scientific publications seems to have stopped increasing trends in the occurrence of SBs. However, it does not prevent that, also similar to the natural sciences and engineering, a more a more or less constant fraction of publications still becomes an SB.

We investigate in this study SBs with $s$ = 5, 10, 15, 20, 25, and 30 with a focus on the SBs with $s$ = 5 and 10. But obviously there are more SBs: those with sleeping times $s$ = 6, 7, 8, 9, . . . and so on. In S4 Table we present the numbers of all publications from $s$ = 1 to $s$ = 20. From $s$ = 5 we speak of Sleeping Beauties, but this is of course a matter of convention. Furthermore, it is important to realize that there will be an overlap between the SBs found with, for instance, $s$ = 5 and those with $s$ = 6. An example shows this. Say a paper has in the twelve years starting with the publication year the following series of citation numbers: 0, 0, 0, 0, 1, 1, 5, 9, 10, 15, 11, 10. This paper will be identified as an SB with $s$ = 5, $c_s$(max) = 1.0, and $c_a$(min) = 5.0 because in the first 5 years after publication $c_s$ is (0+0+0+0+1)/5 = 0.2, and in the following 5 awakening years $c_a$ is (1+5+9+10+15)/5 = 8.0. But this paper will *also* be identified as an SB with $s$ = 6, $c_s$(max) = 1.0, and $c_a$(min) = 5.0 because in the first 6 years after publication $c_s$ is (0 +0+0+0+1+1)/6 = 0.33, and in the following 5 awakening years $c_a$ is (5+9+10+15+11)/5 = 10.0. We leave it to the reader that this paper will also be identified as an SB with $s$ = 7.

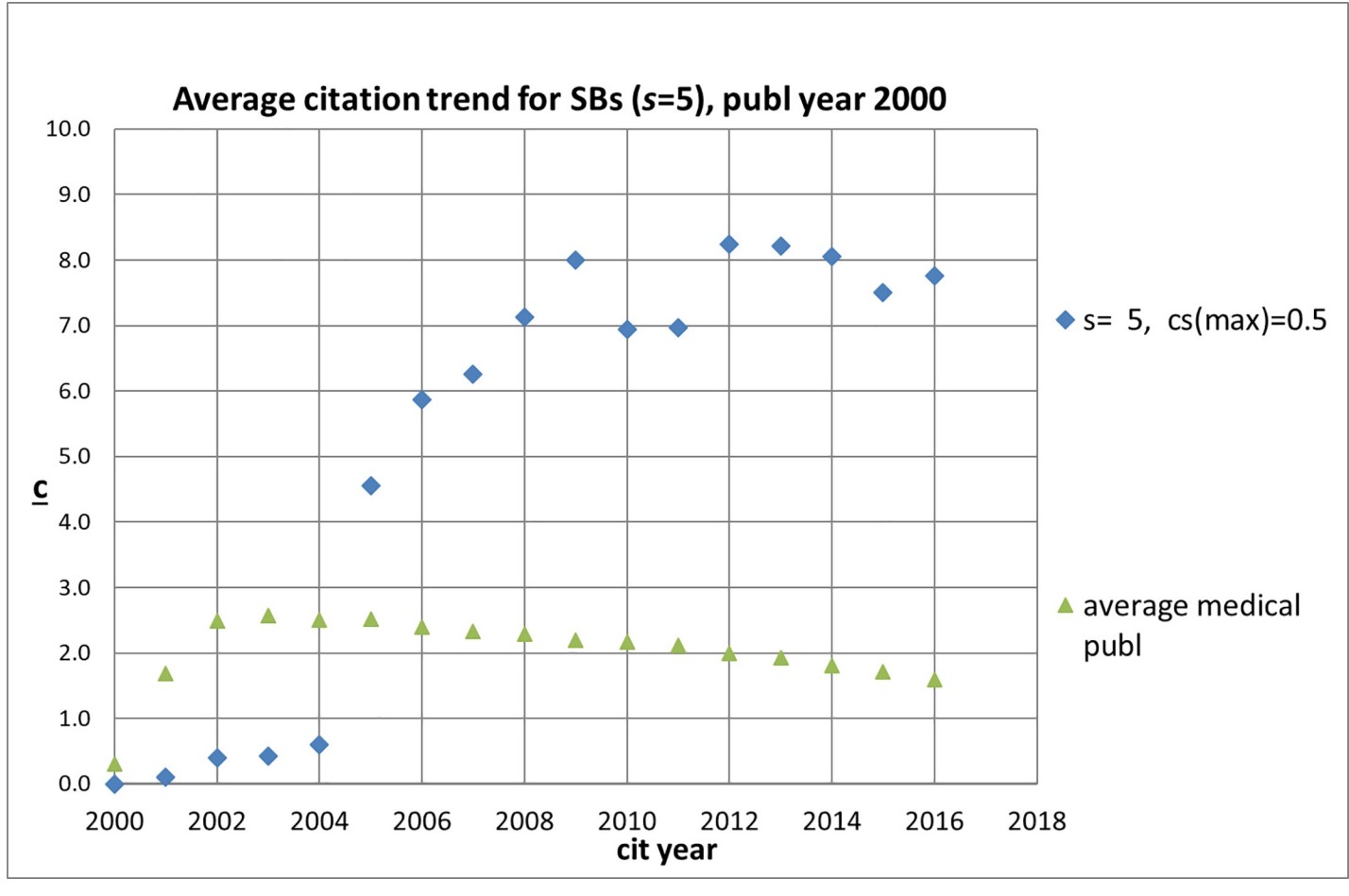

**Fig 4. Average citation ($\underline{c}$) trend in the period 2000–2016 for SBs with $s$ = 5 and $c_s$(max) = 0.5 as well an average medical publication with publication year 2000.**

Therefore the overlap between the SBs with 'multiple' sleeping times has to be determined. We will investigate this in follow-up work more precisely by adding to our search algorithm lower and upper thresholds for the sleeping time, $s$(min) and $s$(max). All in all, we find that the number of unique SBs with sleeping times $s$ = 5, 6, 7, 8, . . . . . . ..18, 19, 20 is around 10,000, thus approximately twice the number of only SBs with $s$ = 5.

## Sleeping beauties appear to be high impact publications

An interesting exercise is to analyze the extent to which SBs are cited on the longer term as compared to an average publication. To that end, we determined for all medical SBs published in 2000 the annual number of citations for the period 2000–2016. Their average citation trend is presented in Fig 4 for $s$ = 5 with a very deep sleep ($c_s$(max) = 0.5, n = 38), and in S1 Fig for $s$ = 5 with a deep sleep ($c_s$(max) = 1.0, n = 343) and for $s$ = 10 also with a deep sleep ($c_s$(max) = 1.0, n = 43) together with the citation trend of an average medical publication. We determined for each SB the total number of citations from 2000 up till now (September 23, 2018). Next, we analyzed where these total numbers of citations are located in the citation distribution of the entire set of all medical research papers published in 2000. We find that of the SBs with $s$ = 5 and $c_s$(max) = 0.5, 55% belongs to the top-10% and all to the top-25% of the citation distribution of all medical research papers. For $s$ = 5 and $c_s$(max) = 1.0 we find practically the same

results: 57% belongs to the top-10% and almost all (99%) to the top-25%. Of the SBs with $s$ = 10 12% belongs to the top-10% and all belong to the top-25%. Therefore, perhaps an unexpected finding is that, in general, Sleeping Beauties appear to be highly cited publications in the longer term. But we also find another interesting phenomenon: the average SB citation trends shows that the awakening process appears to be characterized by a sleep that becomes lighter and lighter, i.e., a slow and small increase of the citation intensity during the sleep period, followed by an abruptly rapid increase of the citation impact.

## Scaling phenomena and grand sleeping beauty equation

In the foregoing section, we presented an overall picture of numbers and trends. We will now go into more detail by investigating the dependence of the number of SBs on the three main variables: sleeping period ($s$), during-sleep citation-intensity ($c_s$), and awakening citation-intensity ($c_a$). In this way, we are able to determine the probability that an SB occurs with specific values of the three above variables. In this sense we go back to our first SB measurements [10] where we used the empirical data to construct a 'Grand Sleeping Beauty Equation' (GSBE).

## Scaling with sleeping period

First, we determine the *number of SBs as a function of sleeping time (s)*. As explained earlier, we have to be careful: for SBs with $s$ = 5 we have publication years in the long measuring period 1980–2007 available for counting. But, for instance, for $s$ = 15 we have the measuring period 1980–1997 and for our extreme case, SBs with $s$ = 30 only 1980–1982 is available as measuring period. Thus, in order to find reliable probabilities of occurrence, we have to take measuring periods which cover the same time period of publication years for SBs with different sleeping times. Moreover, it is very well possible that these probabilities are different for the early 1980s as compared to recent years. In order to investigate this in detail, we apply the following procedure. In a first step we measure the numbers of SBs for the different sleeping periods for all possible publication years (we use successive 5-year blocks of the publication years in order to have a sufficiently large numbers of SBs). The results are shown in Table 3. Because we will compare SBs with *different* sleeping periods in the *same* measuring periods we can use real (not-normalized) numbers.

Next, we examine the relations between the number of SBs for different sleeping periods. Given the very low numbers for the longest sleeping times (see Table 3) we make the analysis for the sleeping periods $s$ = 5, 10 and 15. The results are shown in Fig 5. In order not to

**Table 3. Number of SBs of different sleeping times and for the given range of publication years. In all cases $c_s$(max) = 1.0.**

| s | 1980–1984 | 1981–1985 | 1982–1986 | 1983–1987 | 1984–1988 | 1985–1989 | 1986–1990 | 1987–1991 | 1988–1992 | 1989–1993 | 1990–1994 | 1991–1995 |
|---|---|---|---|---|---|---|---|---|---|---|---|---|
| 5 | 291 | 312 | 344 | 410 | 477 | 591 | 731 | 896 | 970 | 989 | 936 | 863 |
| 10 | 75 | 100 | 129 | 144 | 155 | 143 | 123 | 102 | 100 | 87 | 94 | 107 |
| 15 | 49 | 49 | 47 | 39 | 35 | 41 | 48 | 53 | 57 | 59 | 59 | 62 |
| 20 | 26 | 34 | 36 | 43 | 48 | 52 | 50 | 56 | 56 | | | |
| 25 | 35 | 34 | 43 | 41 | | | | | | | | |
| | 1992–1996 | 1993–1997 | 1994–1998 | 1995–1999 | 1996–2000 | 1997–2001 | 1998–2002 | 1999–2003 | 2000–2004 | 2001–2005 | 2002–2006 | 2003–2007 |
| 5 | 779 | 794 | 866 | 1,043 | 1,246 | 1,428 | 1,568 | 1,660 | 1,662 | 1,568 | 1,449 | 1,342 |
| 10 | 124 | 144 | 178 | 202 | 214 | 212 | 194 | | | | | |
| 15 | 66 | 71 | | | | | | | | | | |
| 20 | | | | | | | | | | | | |
| 25 | | | | | | | | | | | | |

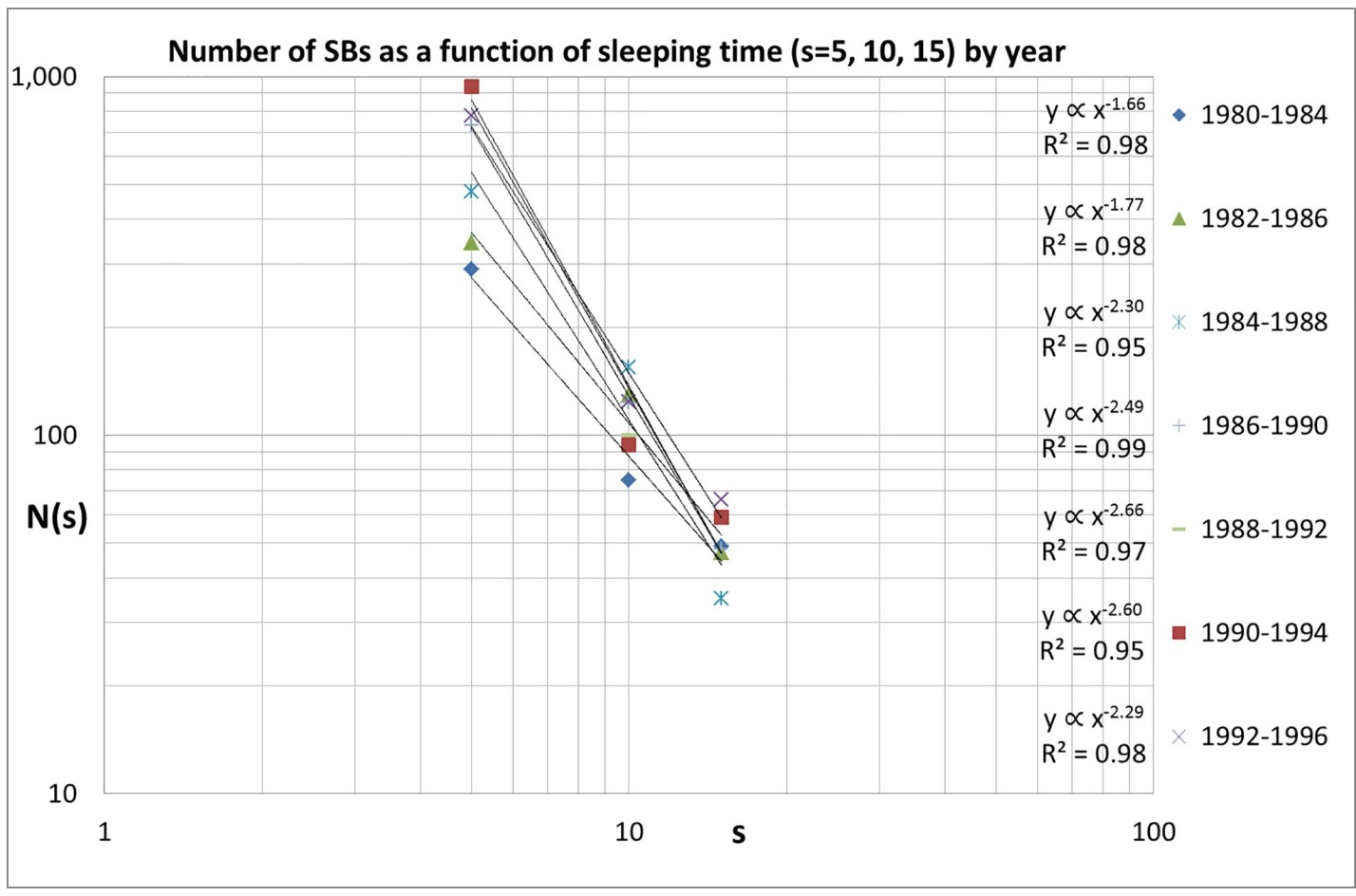

**Fig 5. Number of SBs for *s* = 5, 10, 15 between 1980 and 1996.**

overload the figure we take the data for 1980–1984, 1982–1986, and so on. We find a power law for the scaling of the number of SBs with *s*, with an exponent **α** starting around -1.70 in the early 1980s and a rapid increase to around -2.50.

Finally, the results for the last five publication-year blocks (1994–1998, . . .,1998–2002) covering *s* = 5 and 10 are shown in Fig 6. Again, we see a further continuation of the increasing exponent trend up **α** = -3.02 (1998–2002).

The results are remarkable: the power law scaling of the number of SBs as a function of sleeping time (*s* = 5, 10, 15) increases in the course of time, be it with considerable fluctuations. We find that in the 1980s the scaling of the numbers of SBs as a function of sleeping time has an exponent **α** of around -1.7. At the end of our measuring period however this exponent has almost doubled to around -3.0, see Fig 7. The most probable cause of this change is that the number of SBs with a longer sleeping time decreases. The absolute value of the number of SBs with *s* = 5 influences the power law exponent **α** and therefore the fluctuations in **α** strongly resemble those of the number of SBs with *s* = 5 SBs, as comparison of Fig 3 with Fig 7 shows. For the last publication-year block the number of SBs scales approximately as

$$N(s) = A.s^{-3.0} \qquad\qquad (1)$$

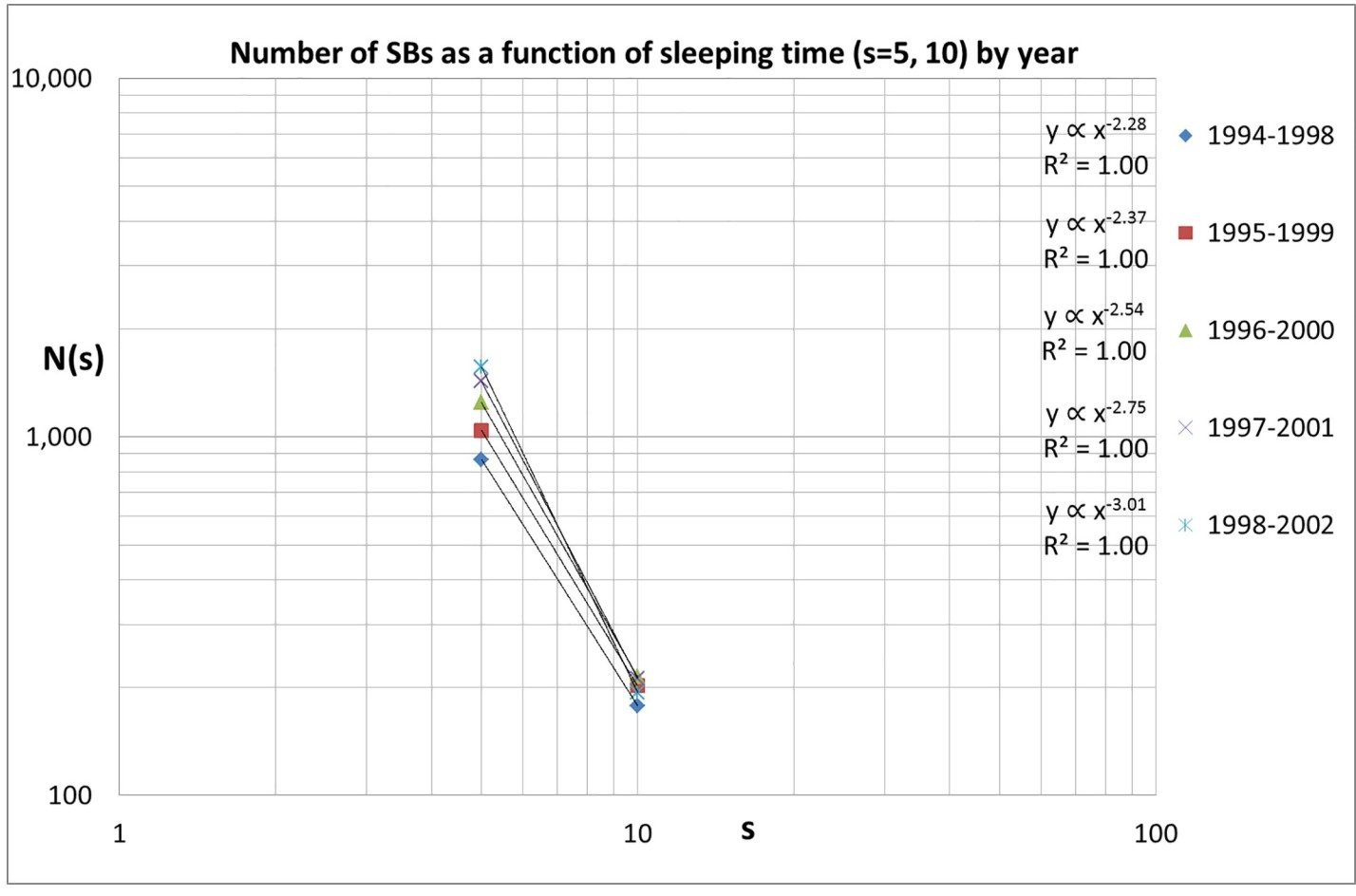

**Fig 6. Numbers of SBs for *s* = 5 and 10, between 1994 and 2002.**

The coefficient **A** is also determined by the empirical results; for instance, for the period 1998–2002 **A** = 200,724. It is interesting to compare these measurements with those in our first Sleeping Beauty paper (hereafter referred to as our 2004-study) [10]. This comparison has its restrictions because this 2004-study concerned all disciplines. We take the same range of publication years as in the 2004-study (1980–1997) and determine for our medical SBs the power law exponent for the relation between the number of SBs with **s** = 5 and 10 with sleeping period. We find $\alpha$ = -2.5 which is in good agreement with the results of the 2004-study where we found a power law scaling with exponent -2.7. Our observations summarized by Eq (1) imply that if a publication is *twice as longer in deep sleep*, the probability that it awakes with, on average, at least five citation per year during five years, is about an *order of magnitude less*. In other words, the longer the sleeping period, the less probable it is that a publication will awake. Indeed a finding that we can intuitively recognize.

## Scaling with sleeping intensity

Next, we analyze the relation between the *number of SBs and sleeping intensity*, i.e., citation-intensity during sleep ($c_s$). This has to be done for each sleeping period separately because SBs with a relatively short sleeping period (**s** = 5) may have different citation-intensity distributions as compared to SBs with a much longer sleeping period (**s** = 10 and longer). We investigated

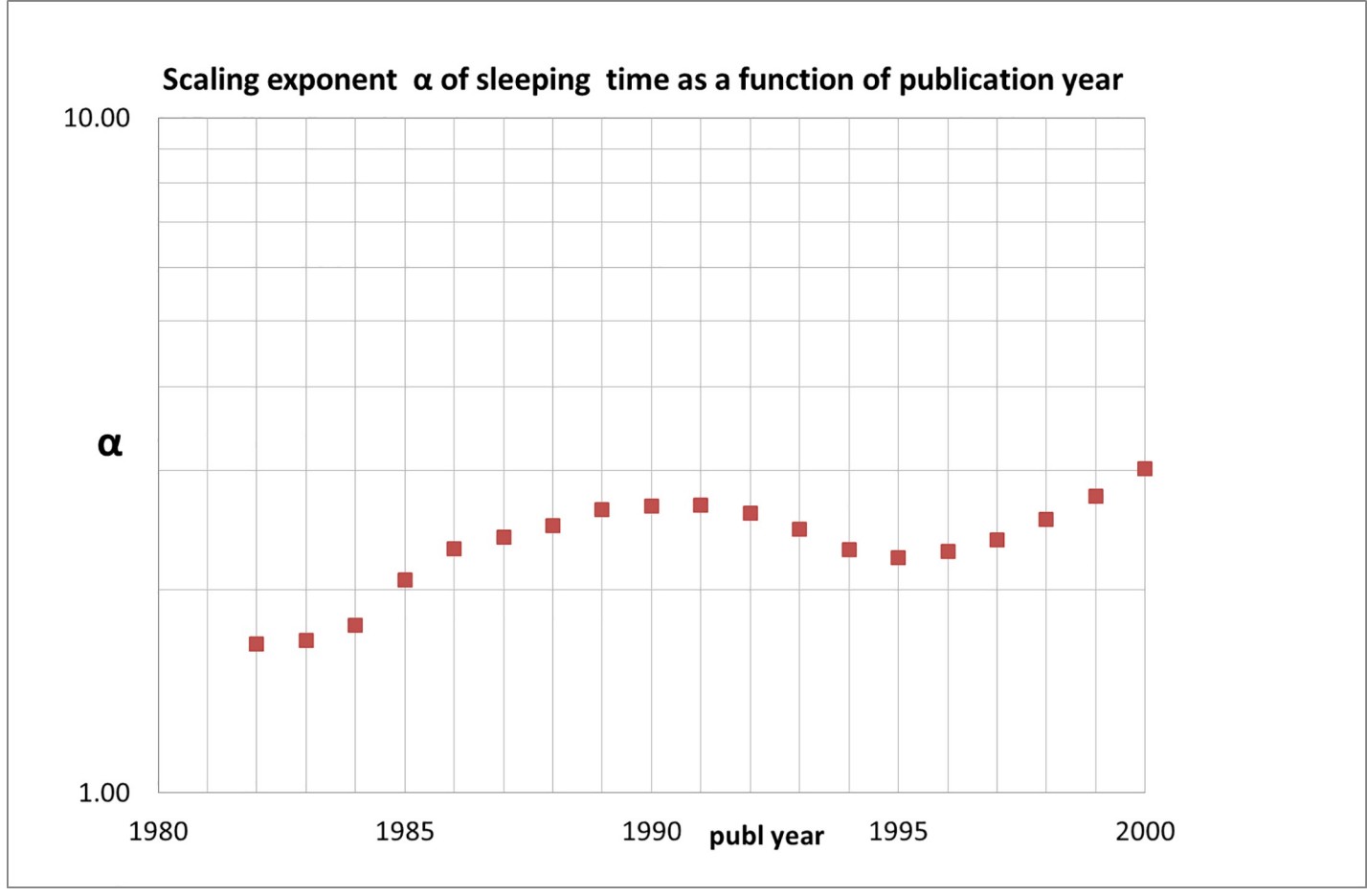

**Fig 7. Scaling exponent α for the whole measuring period 1980–2000 as a function of publication year; numbers are the averages of successive five-years blocks, each five-years block is located in the figure by its middle year (because of the logarithmic scale we take the absolute values of α).**

the scaling of the number of SBs with citation-intensity $c_s$ = 0.2, 0.4, 0.6, 0.8, and 1.0. It is very well possible that citation distributions of SBs will change in the course of time, particularly in a long observation period as used in this study (1980–2017). Therefore, we also analyzed these citation-intensity distributions as a function of the SB publication year. As an example, we show in Fig 8 the results for the SBs with $s$ = 5 for the years 2001–2007. In order to avoid low numbers we again used overlapping 5-years blocks for the publication years. We see that the sleeping-intensity scaling-exponent $\beta$ is around 1.20. In S5 Table we present the results for all years of the measuring period 1980–2007. From these data we deduced the time-dependent development of the sleeping intensity scaling exponent $\beta$, see Fig 9.

Remarkably, scaling exponent $\beta$ fluctuates for a long time -in the period 1980-2000- around +1.40 ($sd$ = 0.08). Then it increases considerably and reaches a value around +1.80 ($sd$ = 0.04) as we see in Fig 9. This implies that the SBs tend to a 'less deep sleep'. Thus, in this recent period the number of SBs scales approximately as (coefficient $B$ follows from the empirical data, for the period 2003–2007 $B$ = 560.37):

$$N(c_s) = B.c_s^{+1.8} \qquad (2)$$

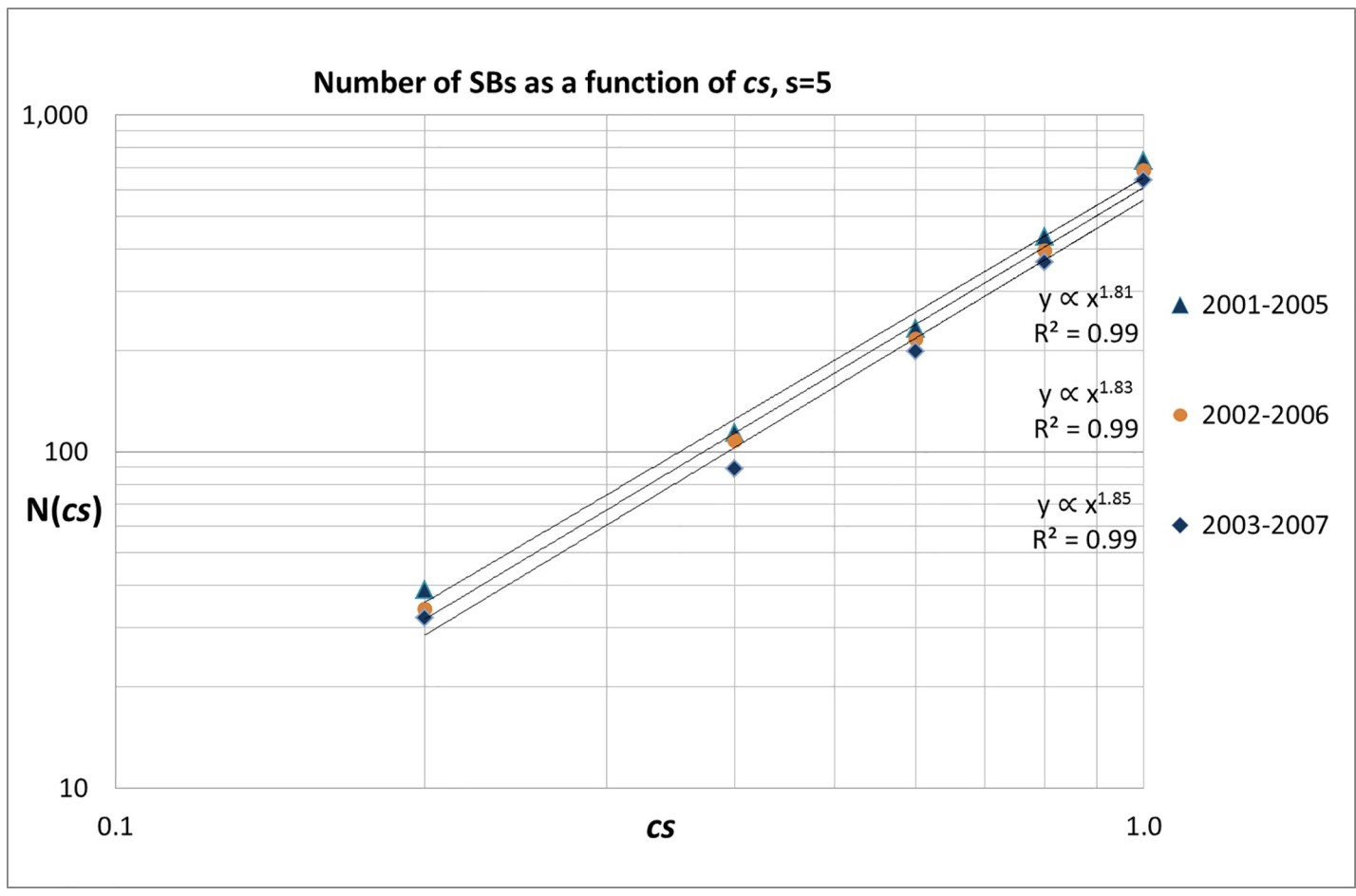

**Fig 8. Number of SBs ($s = 5$) for successive during-sleep citation-intensity ($c_s$) intervals, 2001–2007 (numbers are located with the middle values of the intervals).**

In our 2004-study [10], we found a during-sleep scaling exponent (for all disciplines together) of around +2.5. For SBs with $s = 10$ we find that the during-sleep scaling exponent fluctuates during the whole measuring period (1980–2002) around $\beta = 1.43$ ($sd = 0.18$), see S6 Table and S2 Fig.

Our observations summarized by Eq (2) imply that the probability to find a publication that has twice as much citations in deep sleep (e.g., 4 versus 2 in five years sleep), is about factor 2.5 higher. In other words, the less deep the sleep, the more SBs will be found, because we are moving toward 'normal' publications.

## Scaling with awakening citation-intensity

The third step is to analyze the relation between the number of SBs and awakening citation-intensity ($c_a$). Again, this has to be done for each sleeping period separately because SBs with different sleeping periods may have different citation-intensity distributions during the awakening period. We investigated the scaling of the number of SBs with successive awakening citation-intensity intervals: $5.0 \leq c_a \leq 6.0$, $6.0 < c_a \leq 7.0$, $7.0 < c_a \leq 9.0$, $9.0 < c_a \leq 11.0$, $11.0 < c_a \leq 13.0$, $13.0 < c_a \leq 15.0$, and $15.0 < c_a \leq 17.0$. Also here we analyzed these citation intensity distributions as a function of publication year. Fig 10 shows as an example the results for the SBs with $s = 5$ for the publication years 2001–2007 (divided in three overlapping five-years blocks). We notice

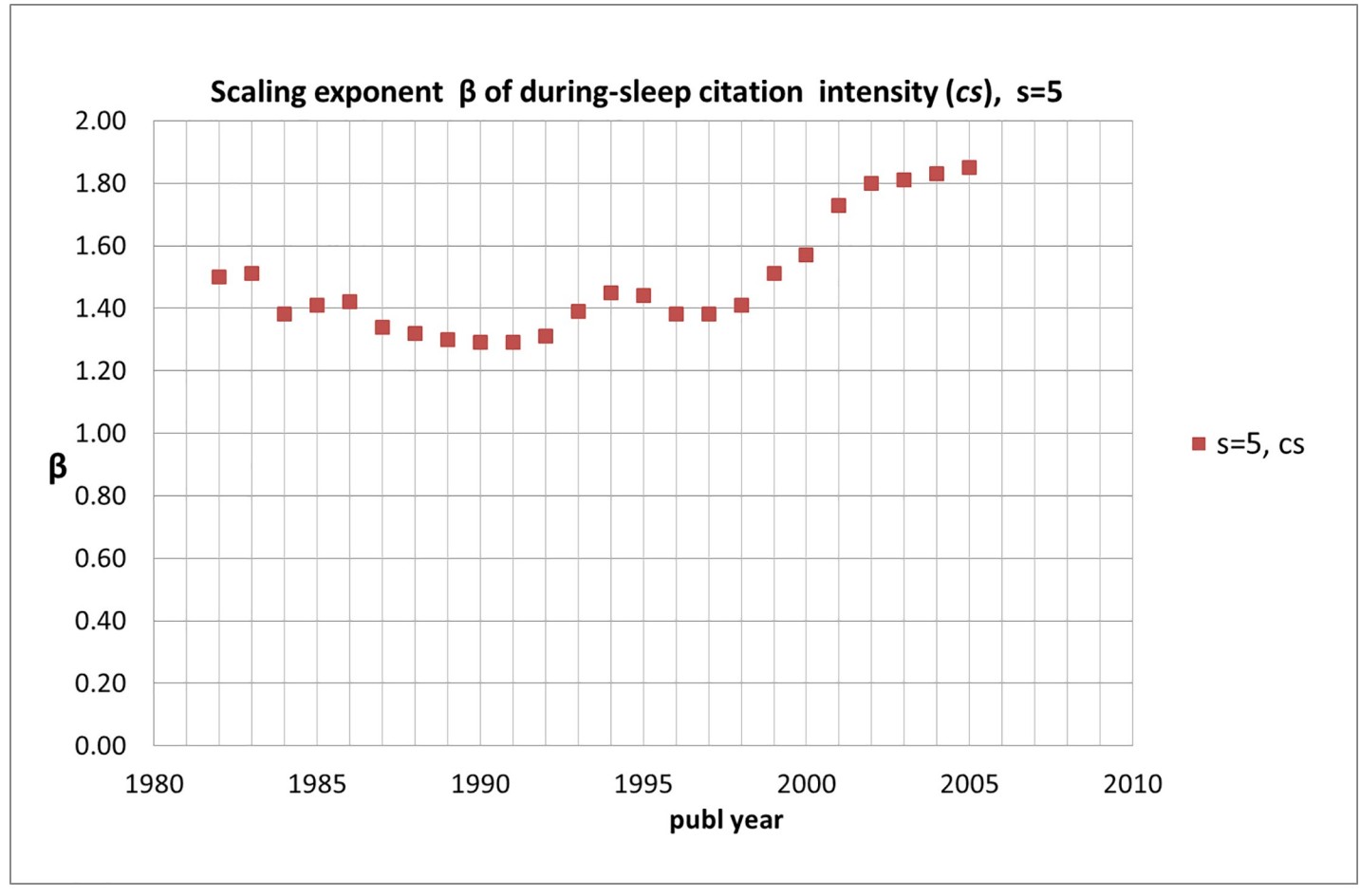

**Fig 9. Scaling exponent β of during-sleep citation intensity, SB-SNPRs with *s* = 5; numbers are the averages of successive five-years blocks, each five-years block is located in the figure by its middle year.**

the very steep decrease of the number of SBs as a function of $c_a$, with exponent γ around -6. As far as we know this is the steepest scaling exponent found in bibliometric research. For instance, in the case of measuring period 2003–2007 the number of SBs scales approximately as:

$$N(c_a) = C.c_a^{-6.1} \qquad (3)$$

Given the steepness of the curve the value of coefficient ***C*** is large, for 2003–2007 it is around $27^*10^6$. In S7 Table we present the results for all publication years in the measuring period 1980–2007. From these data we deduced the time-dependent development of the awakening citation-intensity scaling exponent γ, see Fig 11.

The scaling of the awakening citation-intensity shows an increase starting with γ around -5.0 in the 1980s but since the second half of the 1990s it fluctuates around -6.0. The most plausible explanation for this phenomenon is that in recent times there are relatively less SBs with higher awakening citation-intensity. For SBs with ***s*** = 10 (data in S8 Table) we find that the trend of the awakening citation-intensity exponent γ is rather constant and fluctuates around -4.33 (*sd* = 0.63), see S3 Fig. These values are lower than those in our 2004-study [10] where we found (for all disciplines together) an exponent of -6.6.

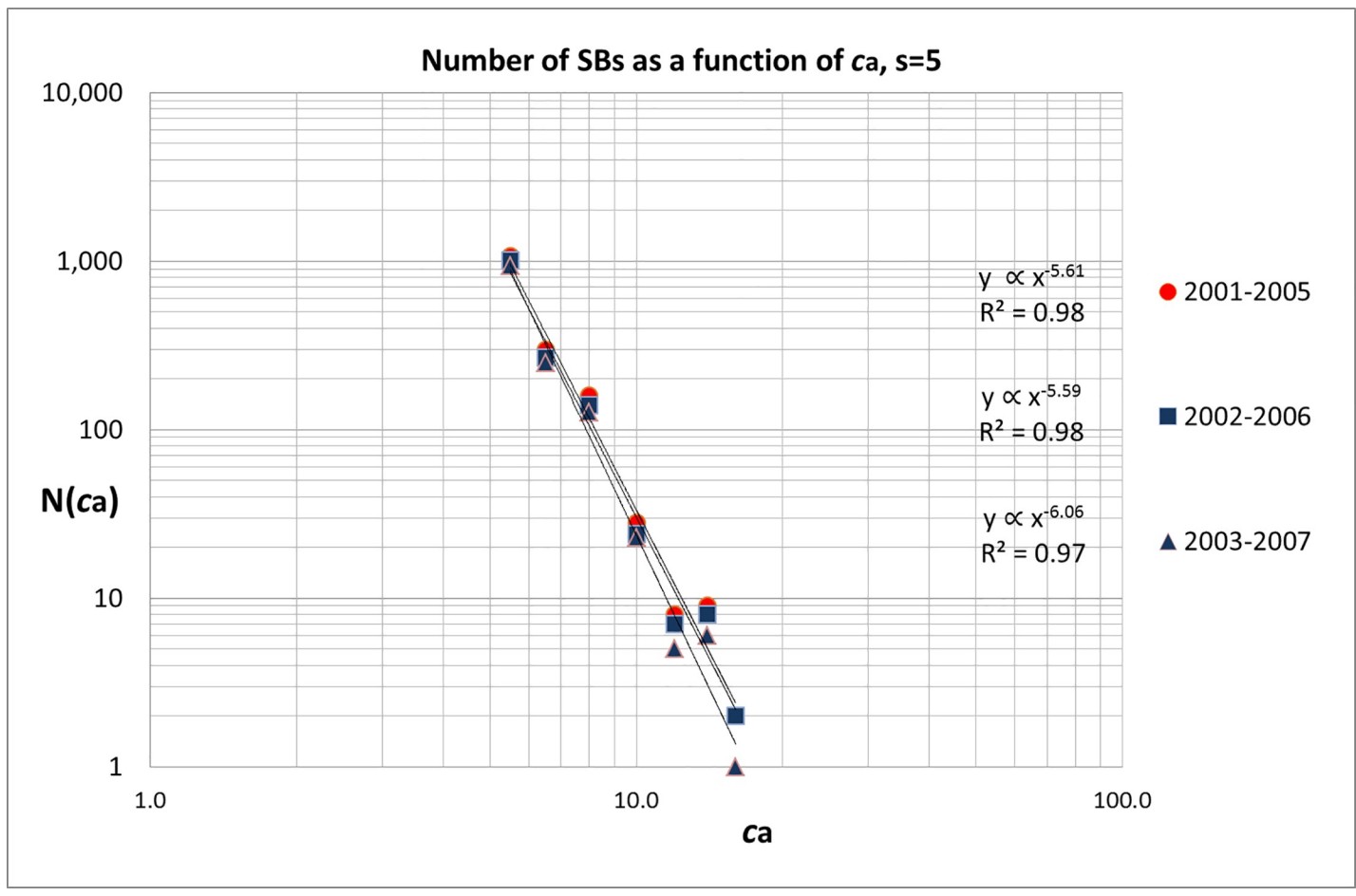

**Fig 10. Number of SBs ($s = 5$) for successive awakening citation-intensity ($c_a$) intervals, 2001–2007.**

All in all these observations summarized by Eq (3) mean that the probability for a higher awakening citation-intensity decreases very rapidly. For instance, the probability that a SB during the awakening period will have a citation intensity twice as large as another SB (e.g., 10 versus 5 citations per year in the five years awakening period), is about a factor 50 lower. Taking the empirical findings for the scaling with sleeping period length and with during-sleep citation-intensity together with the results for the scaling with the awakening citation-intensity, we find for the *Grand Sleeping Beauty Equation* (see van Raan 2004) approximately:

$$N = f(s, c_s, c_a) \propto s^{-3.0}.c_s^{+1.8}.c_a^{-6.1} \qquad (4)$$

Eq (4) gives (after determination of a constant factor given by the coefficients of the above discussed scaling with *s*, *c*$_s$, and *c*$_a$) the number of Sleeping Beauties for any sleeping time, during-sleep intensity and awakening citation-intensity, and particularly the dependency on these variables. Summarizing we find the following main characteristics of Sleeping Beauties in the scientific literature. First, the strong negative exponent connected to sleeping time shows that the probability of awakening after a deep sleep is becoming rapidly smaller for longer sleeping periods. Second, the probability for higher awakening intensities decreases extremely rapidly.

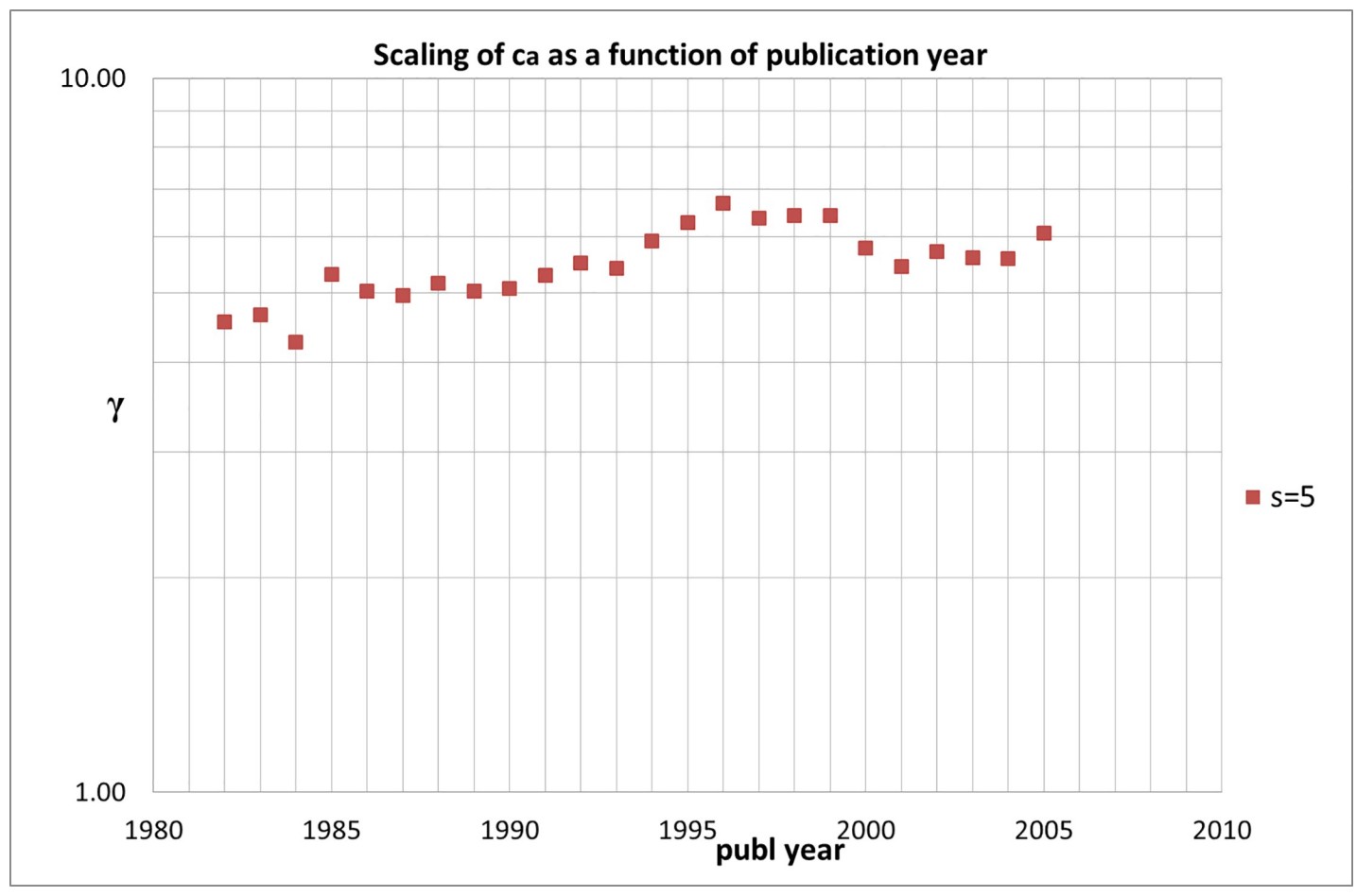

**Fig 11. Scaling exponent γ for the whole measuring period 1980–2007 (because of the logarithmic scale we take the absolute values of γ).**

### Technological impact of sleeping beauties

**Number and distribution of patent citations.** Patents are documents with a legal status to describe and claim technological inventions. Like scientific publications, patents contain references. These references are aimed at proving novelty in view of the existing technological developments. These references mainly relate to earlier patents ('prior art') and to a lesser extent to non-patent items, a major part of which are citations to scientific publications [9], the 'scientific non-patent references' (SNPRs). In the case of scientific publications references are the sole responsibility of the authors, but in patents references can be given by the inventors as well as by the patent examiners. SNPRs represent an important bridge between science and technology [11] but they are not necessarily the direct scientific basis of the invention described in the patent. They are, for instance, used to present a more general scientific basis of the invention. Further research is necessary [12] to analyze the role of SNPRs in relation to the patented invention. Du et al. [13] investigated to what extent patent citations to scientific papers can serve as early signs for predicting delayed recognition. They conclude that non-patent literature cited by patents may provide insight into the awakening of delayed recognition publications.

In this study we elaborate further on our previous work on SBs that are SNPRs (SB-SNPRs) with a particular focus on medical research and on as recent as possible SBs. Patent data were

collected by searching the EPO Worldwide Patent Statistical Database (PATSTAT), Spring 2016 version. We matched all SBs on the basis of their WoS UT-codes with the citations given in patents (for details see [14]) to identify those SBs that are cited by patents. Next, we group patents describing the same invention in 'patent families' [15] to prevent double counting. But even then, it is possible that an SB is matched several times within the same patent family because there may be multiple versions with different patent filing years. Therefore, it is necessary to check for each SB the occurrence of multiple versions and to correct for this. In the case of multiple versions, we take the version with the first filing year. For the sake of brevity, we will speak about patents instead of patent families.

We created a database in which the SBs are uniquely linked with their (first filed) citing patents. This database allows us to determine how many, and which, SBs of sleeping periods ($s$ = 5, 10, 15, 20, 25) are cited by how many, and which, patents within a specified period *after publication*. This is important because, for example, if SBs with $s$ = 5 have citing patents within the first 5 years after their publication, it means that these SBs are cited by patents during their (scientific) sleeping period. Thus, in that case a 'technological prince' is earlier than, or at least at the same time as, the 'scientific prince'. A period of 10 year after publication is the longest period that is applicable to all SBs: the most recent SBs are published in 2007 and a complete 10 years period ends in 2016, the last year of our patent matching measurements. Obviously, for SBs published in 1980 the citation history can be much longer, up to 2016, which means 37 years. Thus, taking the whole available period allows us to study how the technological impact of SBs (based on citations by patents) changed over time. We remind that we analyze SBs with sleeping periods ($s$ = 5, 10, 15, 20, 25) in order find general characteristics. A complete analysis would involve all possible sleeping times $s$ = 6, 7, 8, 9, . . . and so on.

In Table 4 we present the number of SBs cited by patents. This table can be read as follows, we take as an example the SBs with a short ($s$ = 5) sleep. We see that there are in total (1980–2007) 5,261 SBs of which 663 are cited by patents over the whole period (right-most block in the table). Thus, 12.6% of all SBs with $s$ = 5 is an SB-SNPR. These 663 SB-SNPRS are cited in total 2,074 times, by 1,745 patents. With a period of 10 year after publication of an SB, it is found that 492 SBs (9.4% of all SBs with $s$ = 5) are cited 1,279 times by 1,116 patents. In the case of a period of 5 year after publication of an SB with $s$ = 5, all identified patent citations will be *within the sleeping period* of the SBs. We find that 313 SBs are cited within their sleeping period (6.0% of all SBs with $s$ = 5) 565 times by 508 patents. Given that 313 is about 47% of 663, we conclude that almost half of all SB-SNPRs ($s$ = 5) are cited by patents within their sleeping period. As discussed above, the data in Table 4 relate to the total measurement period 1980–2007. In the next section we will analyze the time-dependence of the citations of SBs by patents.

A substantial number of SBs is cited by more than just one patent. Some SBs are cited by more than 10 or even more than 20 patents. It is clear that these highly cited SBs are interesting cases for further analysis, and we will come back later to this issue. In Figs 12 and 13 we present

**Table 4. Number of SBs cited by patents.**

| | SBs | SBs cited | | | | SBs cited | | | | SBs cited | | | |
|---|---|---|---|---|---|---|---|---|---|---|---|---|---|
| | total | within 5y | % | pat cits | pats | within 10y | % | pat cits | pats | up to 2017 | % | pat cits | pats |
| $s$ = 5 | 5,247 | 313 | 6.0% | 565 | 508 | 492 | 9.4% | 1,279 | 1,116 | 663 | 12.6% | 2,074 | 1,745 |
| $s$ = 10 | 614 | 26 | 4.2% | 36 | 34 | 67 | 10.9% | 122 | 114 | 116 | 18.9% | 417 | 331 |
| $s$ = 15 | 199 | 7 | 3.5% | 8 | 7 | 9 | 4.5% | 21 | 19 | 32 | 16.1% | 79 | 74 |
| $s$ = 20 | 110 | 2 | 1.8% | 2 | 2 | 3 | 2.7% | 3 | 3 | 13 | 11.8% | 20 | 18 |
| $s$ = 25 | 62 | 0 | 0.0% | 0 | 0 | 0 | 0.0% | 0 | 0 | 6 | 9.7% | 10 | 10 |

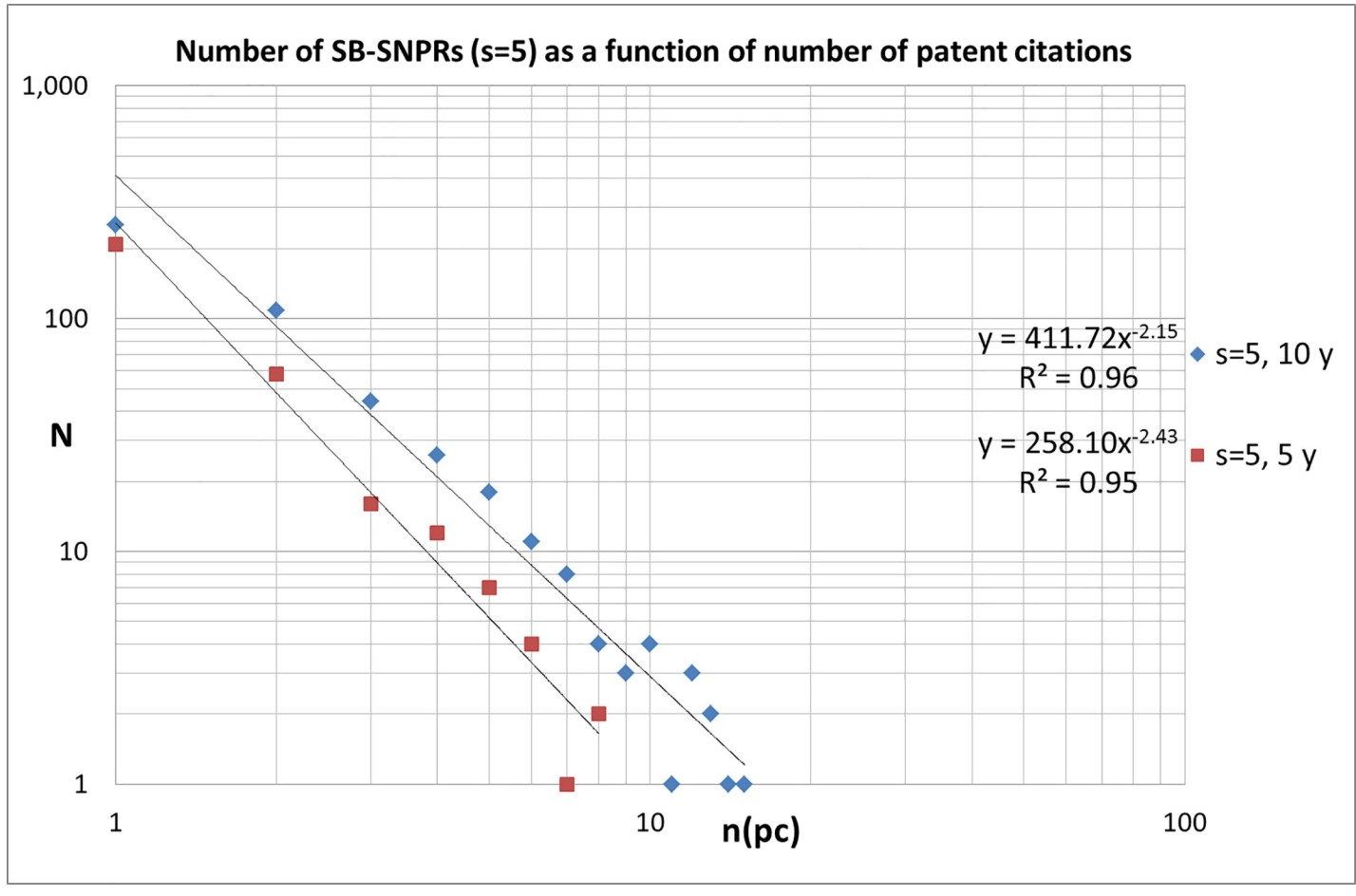

**Fig 12. Number of SBs with *s* = 5 cited by patents as a function of number of patent citations n(pc), citation windows 5 and 10 year.**

the number of SB-SNPRs (Fig 12 for *s* = 5, Fig 13 for *s* = 10) as a function of number of patent citations, with citation windows 5 and 10 year after publication. We see that similar to citations given by publications, also the number of citations given by patents is characterized by a skewed distribution. For our medical SB-SNPRs we find similar distributions as in the case of the natural sciences and engineering [8].

Overall, we find for the SB-SNPRs with the relative short sleep (*s* = 5) that within a period of *10 years after publication* 26% is still not cited (thus they are cited by patents more than 10 years after their publication), about half (54%) is cited by 1 or 2 patents, and about 20% is cited by 3 of more patents. Within a period of *5 years after publication*–which means patent citations *within the sleeping period*- about half (53%) is not cited yet, 41% of the SB-SNPRS is cited by 1 or 2 patents, and 7% is cited by 3 of more patents. For the SB-SNPRs with the long sleep (*s* = 10) we find that within a period of *10 years after publication*–which is in this case *within the sleeping period*- 42% is still not cited, about half (48%) of is cited by 1 or 2 patents, and about 10% is cited by 3 of more patents. Within a period of *5 years after publication*–which means for the SB-SNPRS with *s* = 10 widely within the sleeping period- 78% is not cited yet, 20% of the SB-SNPRS is cited by 1 or 2 patents, and only 2% is cited by 3 of more patents.

For the still longer sleeping periods (*s* = 15, 20, . . .) the numbers are too small for an accurate determination of the distribution. We see in Figs 12 and 13 that the number of SB-SNPRs

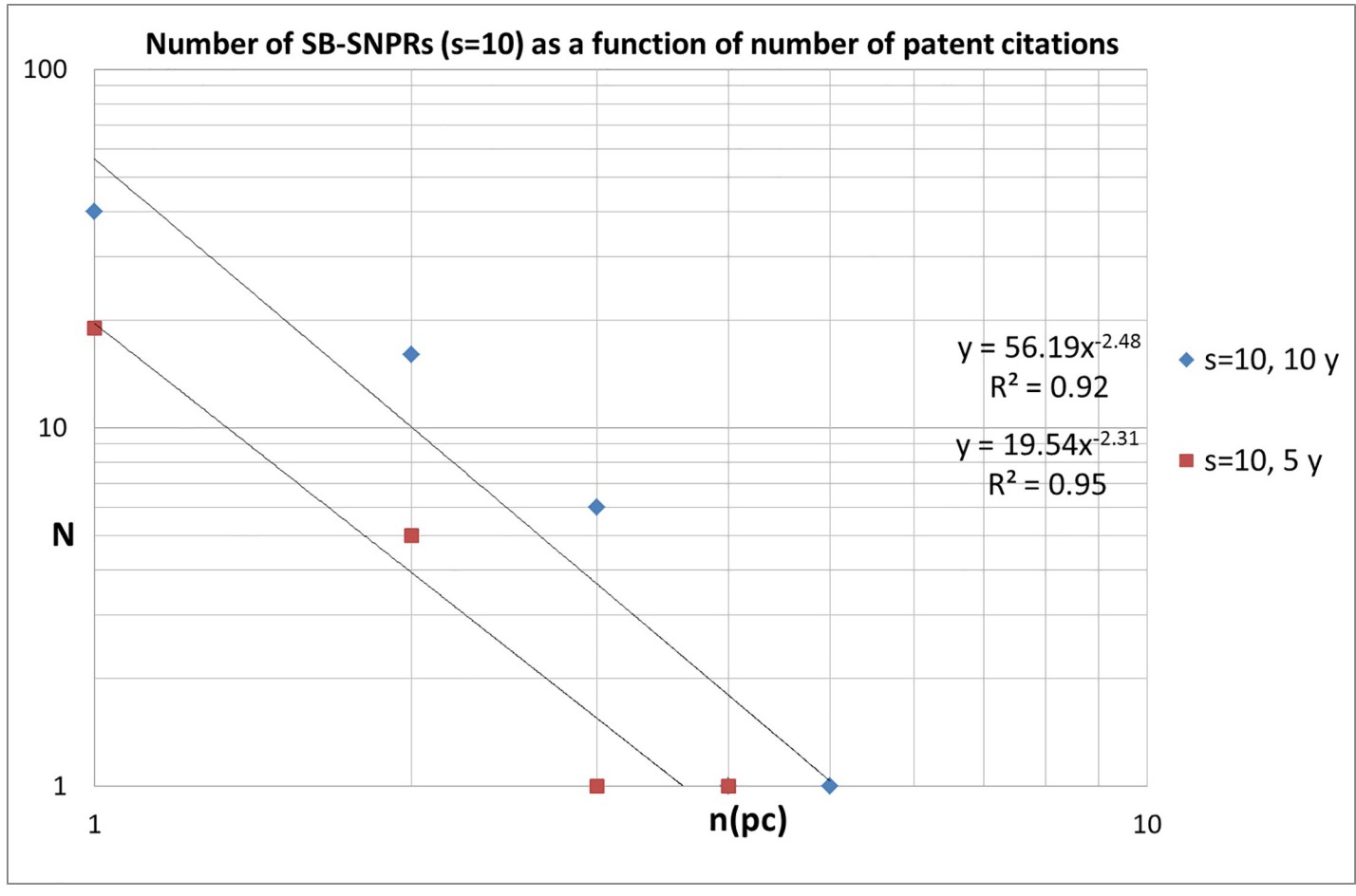

**Fig 13. Number of SBs with _s_ = 10 cited by patents as a function of number of patent citations n(pc), citation windows 5 and 10 year.**

as a function of the number of patent citations scales with a power law exponent between -2.0 and -2.5. For the SB-SNPRs with _s_ = 5 (Fig 12) the distribution for patent citations within 5 years after publication seems to decrease faster than the distribution for patent citations within 10 years. This suggests that it becomes increasingly unlikely to receive more patent citations within 5 years after publication. For the SB-SNPRs with _s_ = 10 (Fig 13) we do not see this effect, but here the numbers are already too small to make a reliable assessment of the details of the distribution.

### Scientific or technological awakening?

The distributions presented in Figs 12 and 13 relate to the total number of SBs in the entire measuring period (1980–1997). As discussed earlier, it is very well possible that properties of SBs and of SB-SNPRs in particular, change over time. This is indeed the case, and quite dramatically: Figs 14 and 15 shows the fraction of SB-SNPRs (Fig 14 for _s_ = 5, Fig 15 for _s_ = 10) that are cited at least once, changes in the course of time. In the 1980s about 50% of the SB-SNPRs with _s_ = 5 and about 20% of the SB-SNPRs with _s_ = 10 were cited within 10 years after publication. But in the first decade of this century almost all SB-SNPRs (both _s_ = 5 as well as 10) are cited within 10 years. For SB-SNPRs with _s_ = 5 the fraction for at least one patent citation within 5 years after publication, and thus within their sleeping period, is in the 1980s

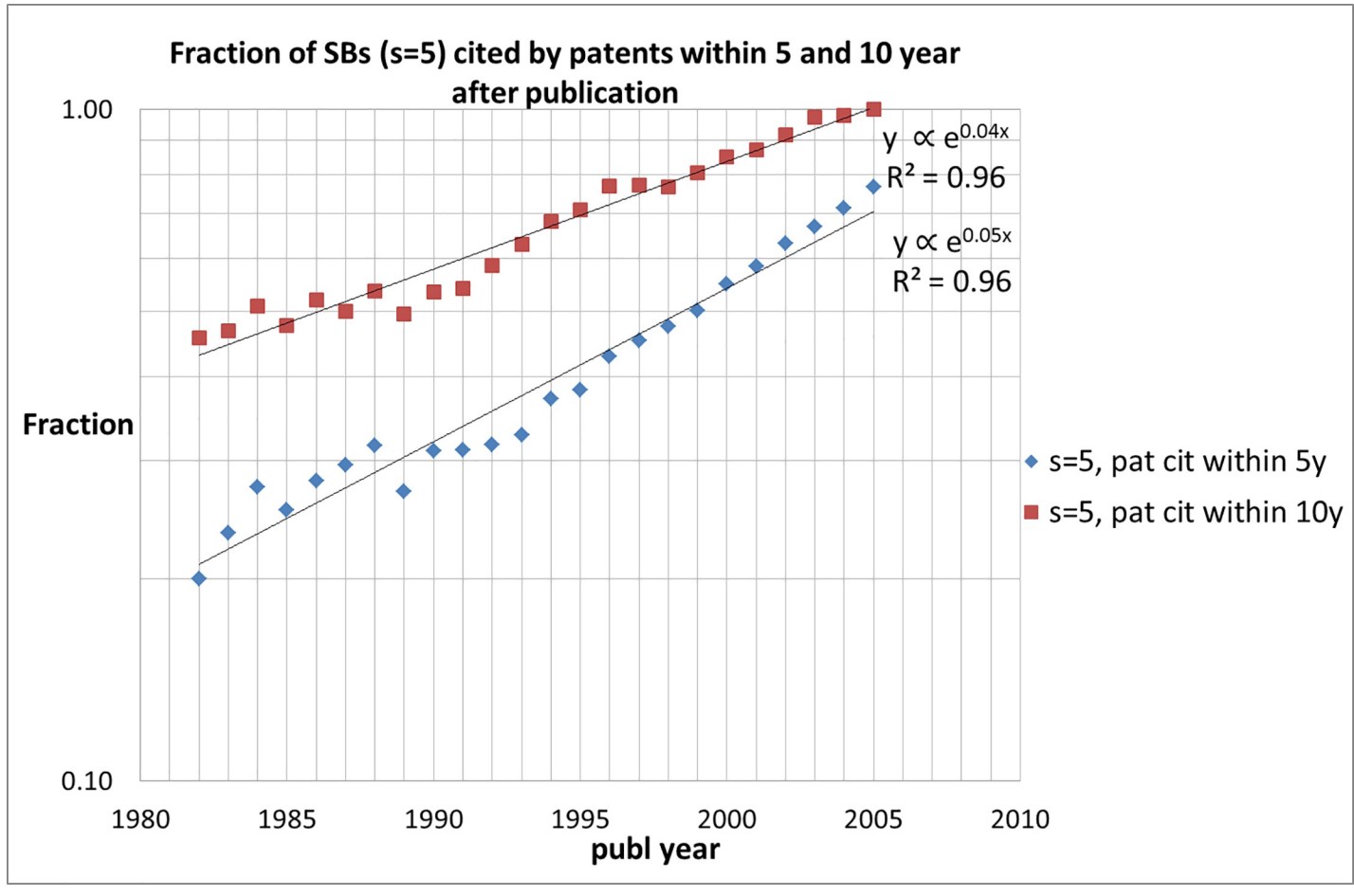

**Fig 14. Fraction of SBs with *s* = 5 that are cited by patents within 5 and 10 year after publication of the SB.**

about 25%. But about 20 years later, in the first decade of this century, the situation has changed drastically: 70–80% of SB-SNPRs are cited within their sleeping period. For SB-SNPRs with the longer sleep period *s* = 10 the fraction for at least one patent citation within 5 years after publication is in the 1980s about 10% and in the first decade of this century it is 40%, again a substantial change. We also notice in Figs 14 (*s* = 5) and 15 (*s* = 10) that the fraction of SBs cited by patents within 5 years increases more rapidly than within 10 years.

All these observations point in one and the same direction: in a rapidly increasing pace publications that still 'sleep' scientifically do already have a technological impact within their sleeping period. We must however be cautious about concluding that a technological awakening is more and more likely to occur than a scientific one. The average SB citation trends (Fig 4) show that the awakening process appears to be characterized by a sleep that becomes lighter and lighter, i.e., a slow and small increase of the citation intensity during the sleep period (but still meeting the requirement $c_s(\max) = 1.0$), followed by an abruptly rapid increase of the citation impact. What we can conclude is that technological impact of SBs in the form of citations by patents will take place in an increasingly shorter period of time after publication, and more and more in the 'formal' sleep period of an SB.

To investigate the increasingly faster pace of technological impact in more detail, we analyzed the time lag between the filing year of the patent that is the *first* citer of the SB-SNPR (earliest filing year) and the publication year of the SB-SNPR (*fpcy*). This *technological time lag*

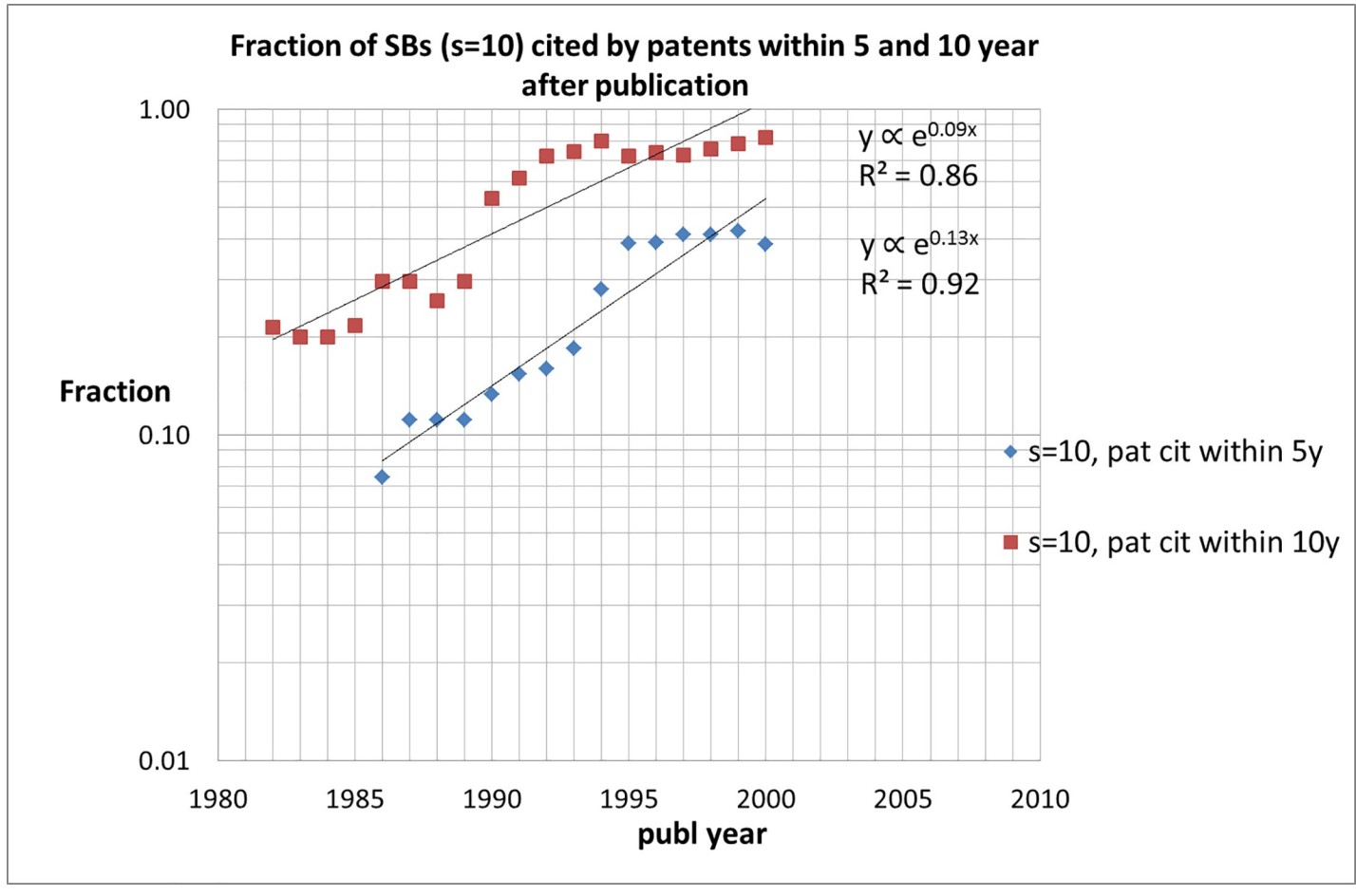

**Fig 15. Fraction of SBs with _s_ = 10 that are cited by patents within 5 and 10 year after publication of the SB.**

ranges for the SB-SNPRs from 1 to 30 years. We calculated averages of **_fpcy_** for successive, partly overlapping 5-year periods: 1980–1994, 1981–1985, . . ., 2004–2007. Fig 16 shows the results. We see that for SB-SNPRs with **_s_** = 5 the technological time lag remains rather stable in the 1980s but after 1990 it becomes rapidly shorter (with about 7% per year) and around 2002 it becomes shorter than the sleeping time. We notice that this trend of the technological time lag does not differ that much for the SB-SNPRs with **_s_** = 5 and **_s_** = 10. The technological time lag becomes already in the early 1990s shorter than the sleeping time for the SB-SNPRs with **_s_** = 10.

Just like our findings for the natural sciences and engineering these observations suggest that in the more recent years the probability that SBs are cited in a patent during sleeping time is increasing. This phenomenon is most related to two important developments: the increasing number of patents and the increasing number of patent citations to scientific publications. We intend to focus on these developments in our follow-up research. In the next section we discuss our first findings.

## Trends in patent scientific intensity

It hardly leads to doubt that the increase since 1980 of the number of SB-SNPRs that are cited by patents within 5 (for SBs with **_s_** = 5) and 10 year (for SBs with **_s_** = 10) after publication of

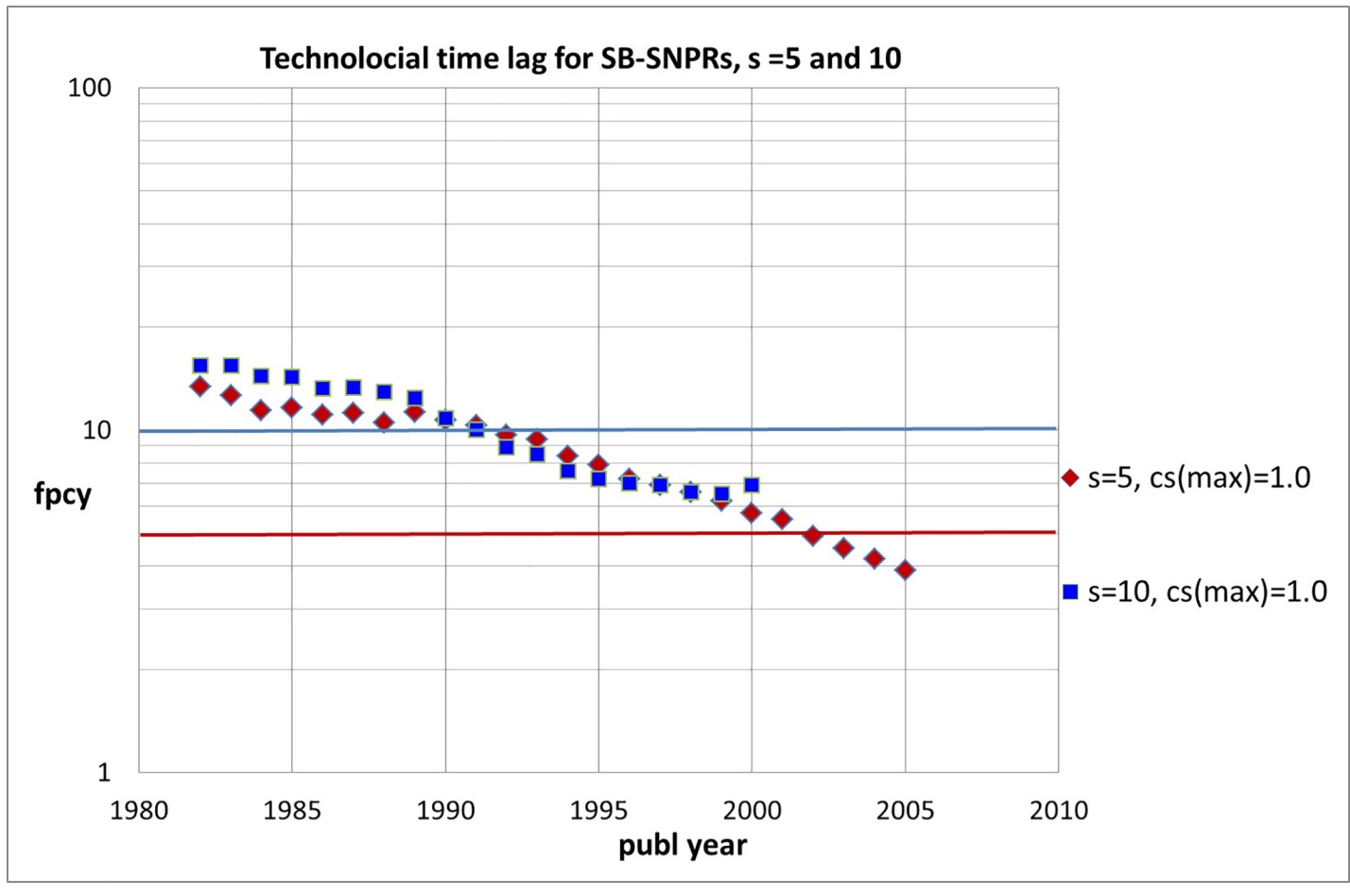

**Fig 16. Technological time lag for SB-SNPRs with $s$ = 5 (red diamonds) and $s$ = 10 (blue squares). In both cases $c_s$ (max) = 1.0. The publication years on the abscissa are the middle years of the successive 5-years periods. The bold horizontal lines mark the lags (*fpcy*) of 5 (red line) and 10 years (blue line) which corresponds to the sleeping time for the SB-SNPRs with $s$ = 5 and $s$ = 10, respectively.**

the SB is influenced by the increasing number of patents as well as the increasing science intensity of patents, i.e., the share of patents citing scientific publications. For the sake of clarity, we re-emphasize that these publications cited by patents are indicated by SNPRs, and if these publications are Sleeping Beauties we indicate them by SB-SNPRs.

To get more insight into the above discussed developments we first calculated several important general trends in patents. In Fig 17 we show the trend of patents [16] as well as the trend of the number of patents that cite one or more scientific publications (i.e., patent with at least one SNPR). These trends are based on those patents with an EP (European Patent Office) or WO (World Intellectual Property Organization) patent publication but *also* a US patent publication. We think that these patents relate to inventions that are expected to be worth the most.

As we notice in Fig 17, data given more recently than 2015 are not shown. This is because for the recent years the patent database is not yet complete. The decrease from 2012 of the patents with SNPRs is probably caused by delays in citing SNPRs which may take several years. The percentage of patents with SNPRs, which we regard as discussed earlier as a simple indicator of patent science intensity, increased in the period 1980–1995 and becomes more or less stable since 1995 at around 20%. It is quite remarkable that the science intensity of patents

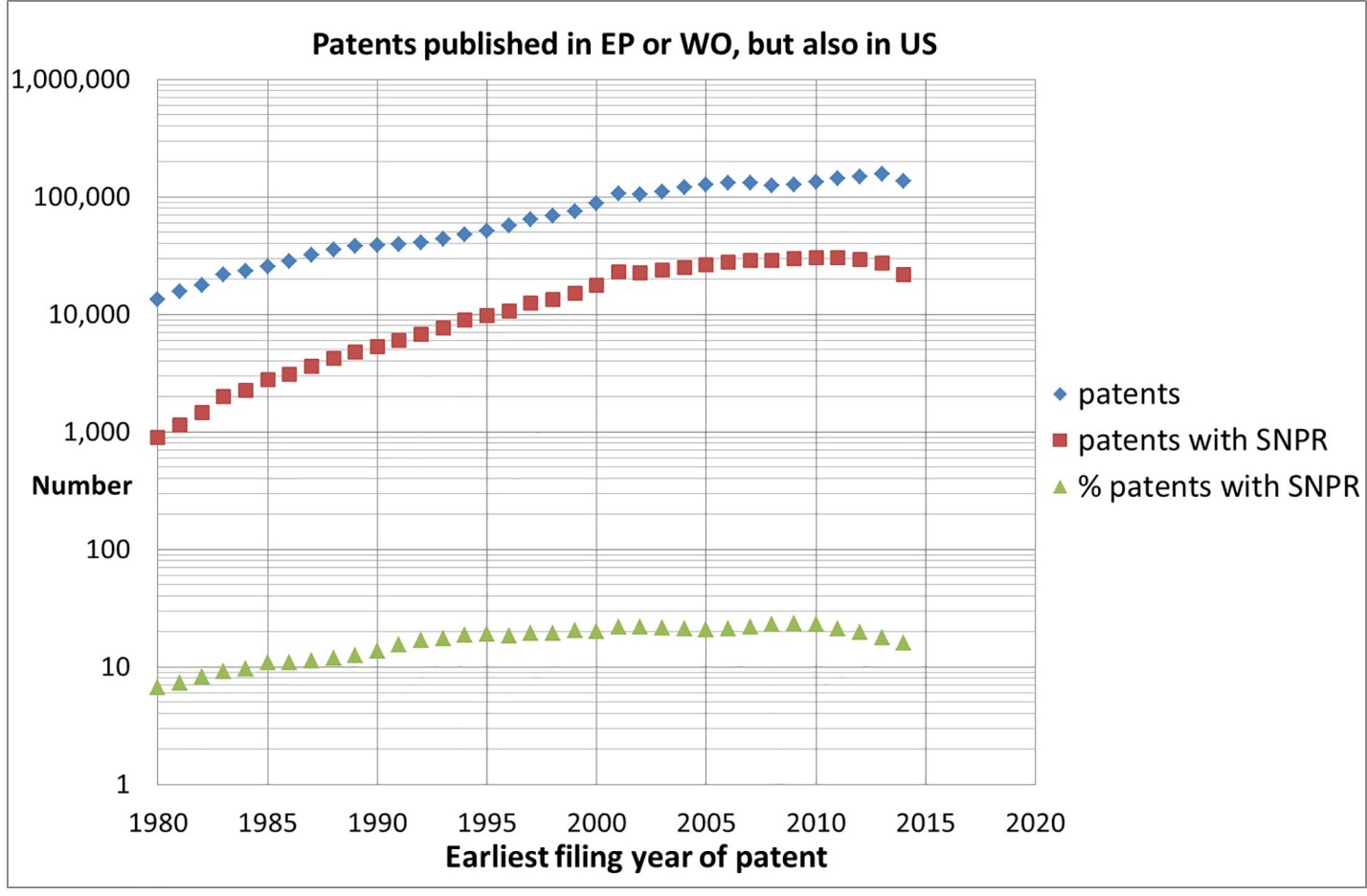

**Fig 17. Trend of the total number of patents with at least a patent publication in EP or WO but also in the US; the trend of patents with SNPRs, as well as the percentage of these patents.**

remained stable in the last nearly 25 years. The decline in the most recent years is most probably again a database effect caused by the complexity of the patent procedures particularly the inclusion of references.

Next, we focus on the science intensity of technological fields that are relevant for our medical SB-SNPRs. In order to find these technological fields (defined by International Patent Classification (IPC) codes [17, 18]), we determined the following variables: (a) the number of SB-SNPRs cited by patents in a specific technology field; (b) the share of technological fields in the total number of medical SB-SNPRs; (c) the share of a technology field in the total number of patents that have SNPRs (thus all SNPRs, not only the SB-SNPRs); and (d) the share of a technology field in all patents. We assume that variable (b) is the most interesting one for this study. We then calculated a composite indicator called 'science intensity index' by multiplying variable (b) (the share of technological fields in the total number of medical SB-SNPRs, we consider this as a weight of a technology field for the medical SB-SNPRs) with variable (c) (the share of a technology field in the total number of patents that have SNPR).

In Fig 18 we show the trend of the science intensity index (1990–2012, years relate to the earliest filing year in a patent family [19]). for the six technological fields that are the most important for the medical SB-SNPRs: Pharmaceuticals, Medical Technology, Biotechnology, Organic Fine Chemistry, Analysis of Biological Materials, and Basic Materials Chemistry. We

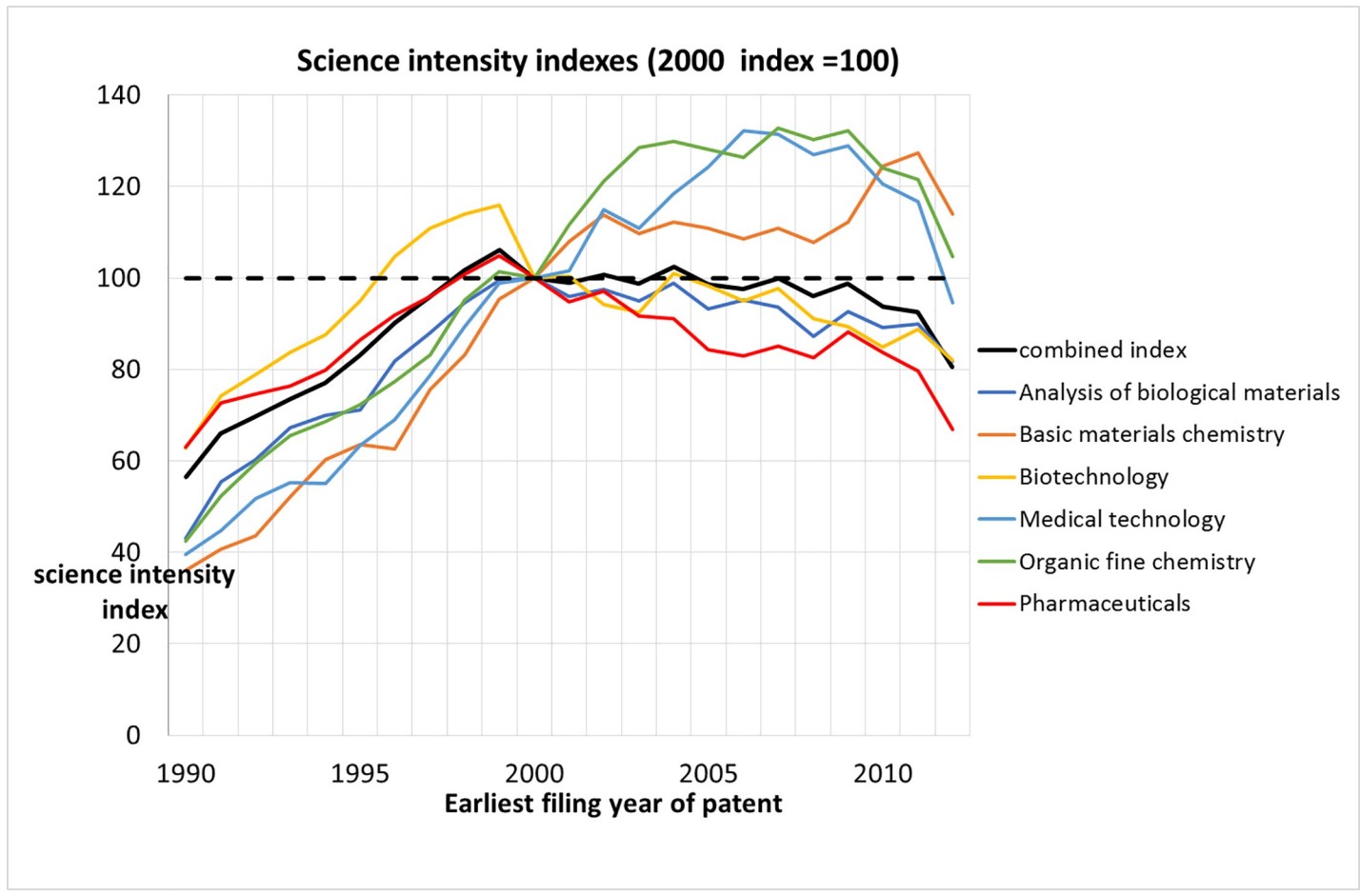

**Fig 18. Trend of the composite indicator for six technological fields that are the most important for medical SB-SNPRs.** The indicator value in the year 2000 is taken as a reference value (index = 100) for each field.

take 2000 as the index year (value = 100). The science intensity index increases between 1990 and 2000, but from 2000 the six fields develop quite differently. However, these measurements must be improved by accurately considering the delay in patent citations to scientific publications. This will be the subject of a follow-up study and therefore proper normalization of the number of SB-SNPRs on the basis of the science intensity is in the context of this study not possible.

### Influence on the technological time lag by inventor-author self-citation

In foregoing studies [7, 8] focusing on physics, chemistry, engineering and computer science, we investigated the extent to which at least one of the *inventors of patents citing* SB-SNPRs is also an *author of the cited SB-SNPR*. We call this inventor-author self-citation [20]. We concluded that only for a small minority (5%) of the SB-SNPRs the authors are also inventors of the technology described in the citing patent. In this study we find that for the 663 SB-SNPRs with *s* = 5 inventor-author self-citation occurs in the case of 59 SB-SNPRs, i.e., 9%. These 59 SB-SNPRs are cited by 206 patents. In nearly half of these 206 patents, 95, we have inventor-author self-citation. This occurrence of inventor-author self-citation is comparable though somewhat higher as compared to our findings for physics, chemistry, engineering and computer science.

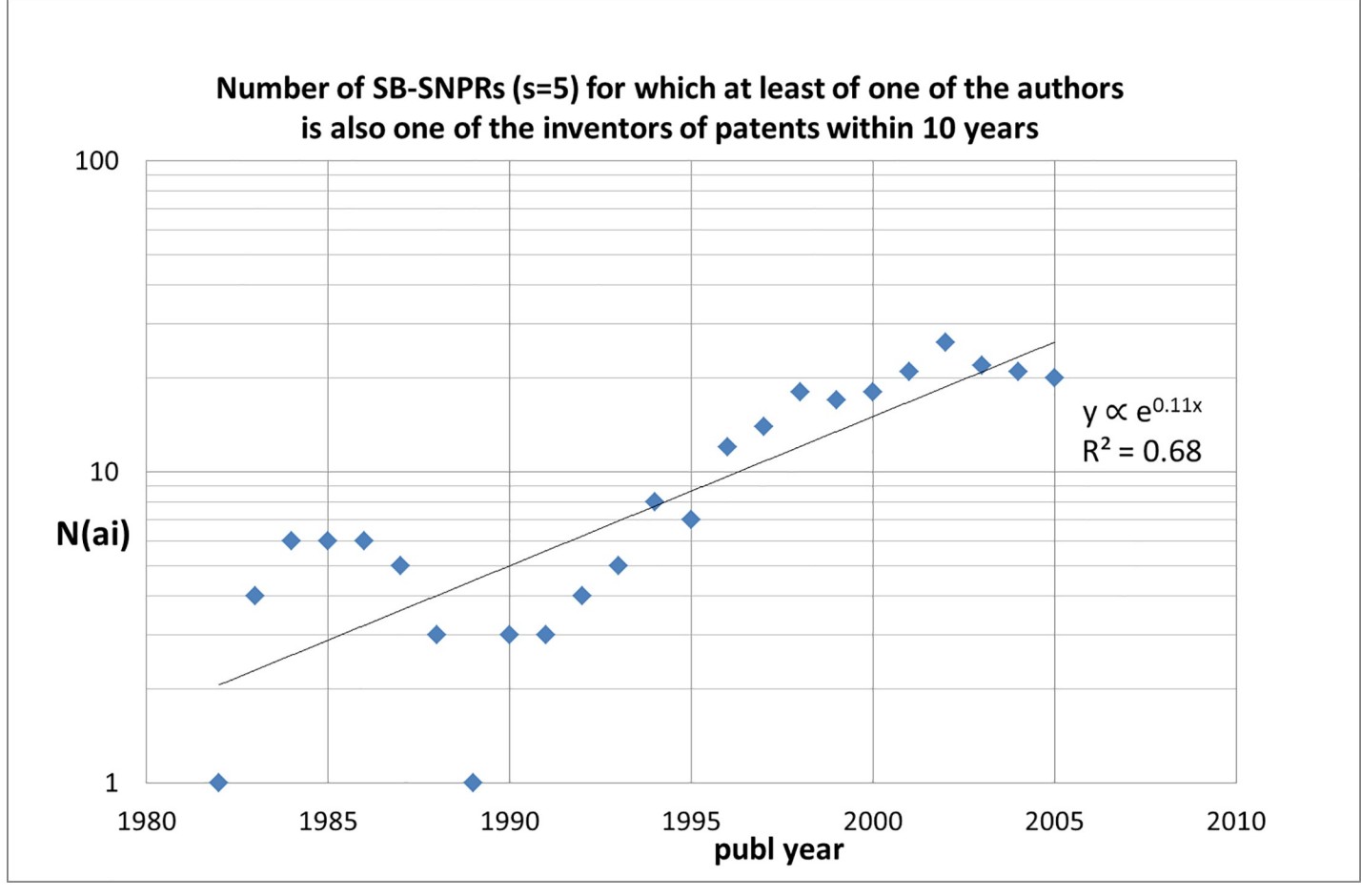

**Fig 19. Number of SB-SNPRs (*s* = 5) of which at least one of the authors is also one of the inventors of the citing patent (N(ai)) as a function of the publication year of the SB-SNPRs.**

A new element in this study is the time trend of inventor-author self-citations. In Fig 19 we show for successive five-year time blocks the number of SB-SNPRs with *s* = 5 of which at least one of the authors is also one of the inventors of the citing patent. This number increases with about 11% per year: in recent years the occurrence of inventor-author self-citations is about four to five times higher than in the 1980s.

Are inventor-author self-citations responsible for earlier first patent citations of the Sleeping Beauties and thus shorter technological time lags? We used the analysis of the technological time lag discussed in the forgoing section to investigate this. For SB-SNPRs with *s* = 5 we removed in our analysis from 1990 the patents that are the first citers in the case of inventor-author self-citation (inventor-author correction). The results are shown in Fig 20. The data indicated with the blue triangles are the corrected data. Without corrections for inventor-author self-citation the results are the same as in Fig 14 (from 1990). We see that inventor-author self-citations indeed result in shorter technological time lags, but this effect is small. This finding confirms the results of our earlier studies that inventor-author self-citation is quite rare. In these earlier studies we also found on the basis of a number of individual cases earlier *scientific* awakening triggered by inventor-author self-citation is small. In this study we find that earlier *technological* awakening triggered by inventor-author self-citation is also small.

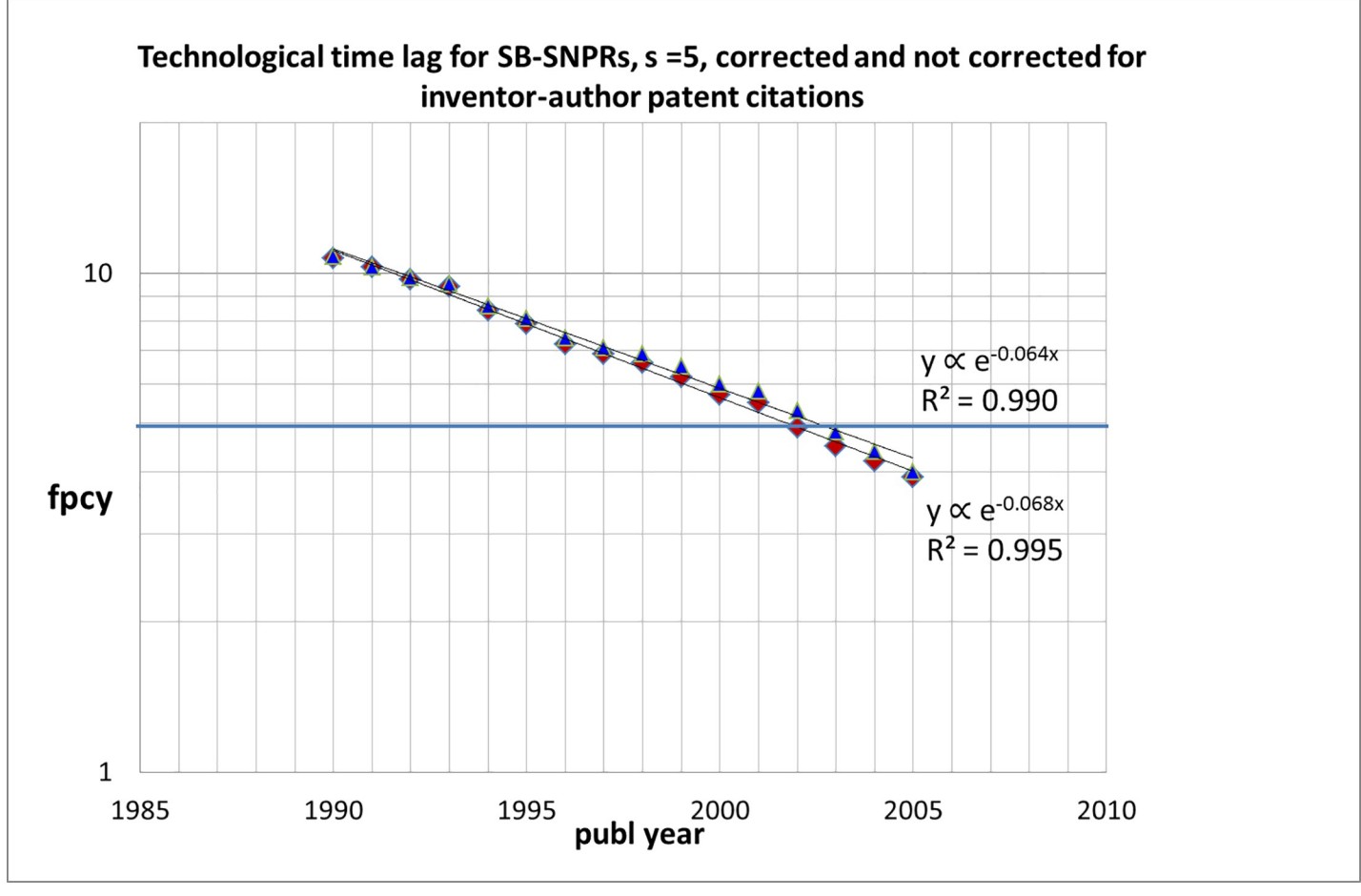

**Fig 20. Technological time lag (fpcy) for SB-SNPRs with _s_ = 5 from 1990: corrected (blue triangles) and not-corrected (red diamonds) for inventor-author patent citations.** The years indicated on the abscissa are the middle years of the successive 5-years periods. The bold horizontal line indicates a time lag of 5 years which is the sleeping time for the SB-SNPRs.

## Scientific and technological impact

The impact of patents on technological development varies considerably, only a small number of patents represents important technological breakthroughs [21]. Patent-to-patent citations provide a first indication of the importance of patents is given by patent-to-patent citations [9]. This is particularly the case if patents become highly cited [22, 23]. We first analyzed for all patents citing the SB-SNPRs the number of times they themselves are cited by other patents. In order to analyze a time trend and to compare the successive years within our measuring period 1980–2007, we have to work with fixed citation windows. First, we select all patents that cite SB-SNPRs (_s_ = 5) _within 10 years after the publication of the SB_. Second, for these patents we counted the citations they receive from other patents _within 5 year after of the filing date of the cited patent_. By ranking these patents by the number of their patent-citations we were able to identify the top-20% cited patents. Finally, we determine the number of SB-SNPRs that are cited by one or more of these top-20% patents. We find that of the 663 SB-SNPRs with _s_ = 5, 107 (16%) are cited by one of more of the 381 top-20% highly cited patents. The last SB publication year we can take into account is 2002: we need a 10-year period for the SBs to be cited by a patent after their publication, and subsequently 5 years for these patents to be cited by other patents, so in total 15 years.

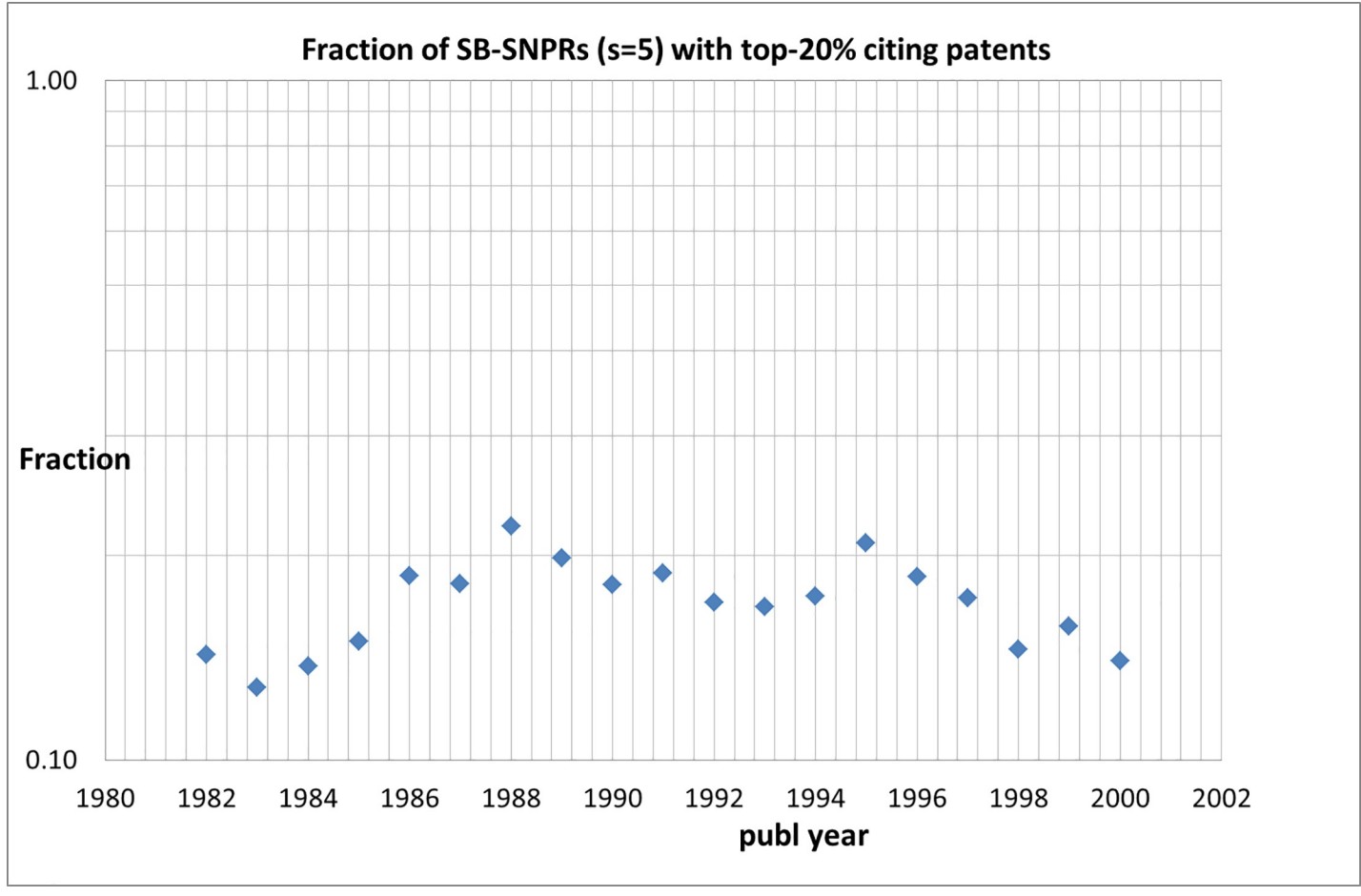

**Fig 21. Fraction of SB-SNPRs (*s* = 5) that are cited by top-20% patents.**

The last complete year of our patent database is 2015 and thus the last SB publication year must be 2002. Fig 21 shows that, remarkably, the fraction of these SB-SNPRs with citing patents belonging to the top-20% remains more or less constant around 0.16.

For both the SB-SNPRs as well as the SB-nonSNPRs with *s* = 5 we found no significant correlation between the number of citations by other publications during the sleeping period (measured by $c_s$) and the number of citations by other publications during the awakening period (measured by $c_a$). In other words, similar to our earlier findings for the natural sciences and engineering, the depth of the sleep is no predictor for the awakening intensity. Also, we did not find a significant correlation between the awakening intensity and the impact of patents that cite the SB-SNPR, i.e., the number of time these patents are cited themselves by other patents within ten years after the filing date of the patent. This means that the scientific impact of Sleeping Beauties is generally not related to the technological importance of the SBs, as far as measured with number and impact of the citing patents. Again, this is similar to our earlier findings for the natural sciences and engineering. Remarkably, however, we do find that the average number of citations (by other publications) during the awakening period ($c_a$) tends to be higher for the SB-SNPRs (*s* = 5), but this difference disappears gradually, see Fig 22.

In our set of 5,247 SBs with *s* = 5 one SB, also an SB-SNPR, immediately strikes the eye because of an enormous increase of citations: Matthews et al (1985, University of Oxford) [24],

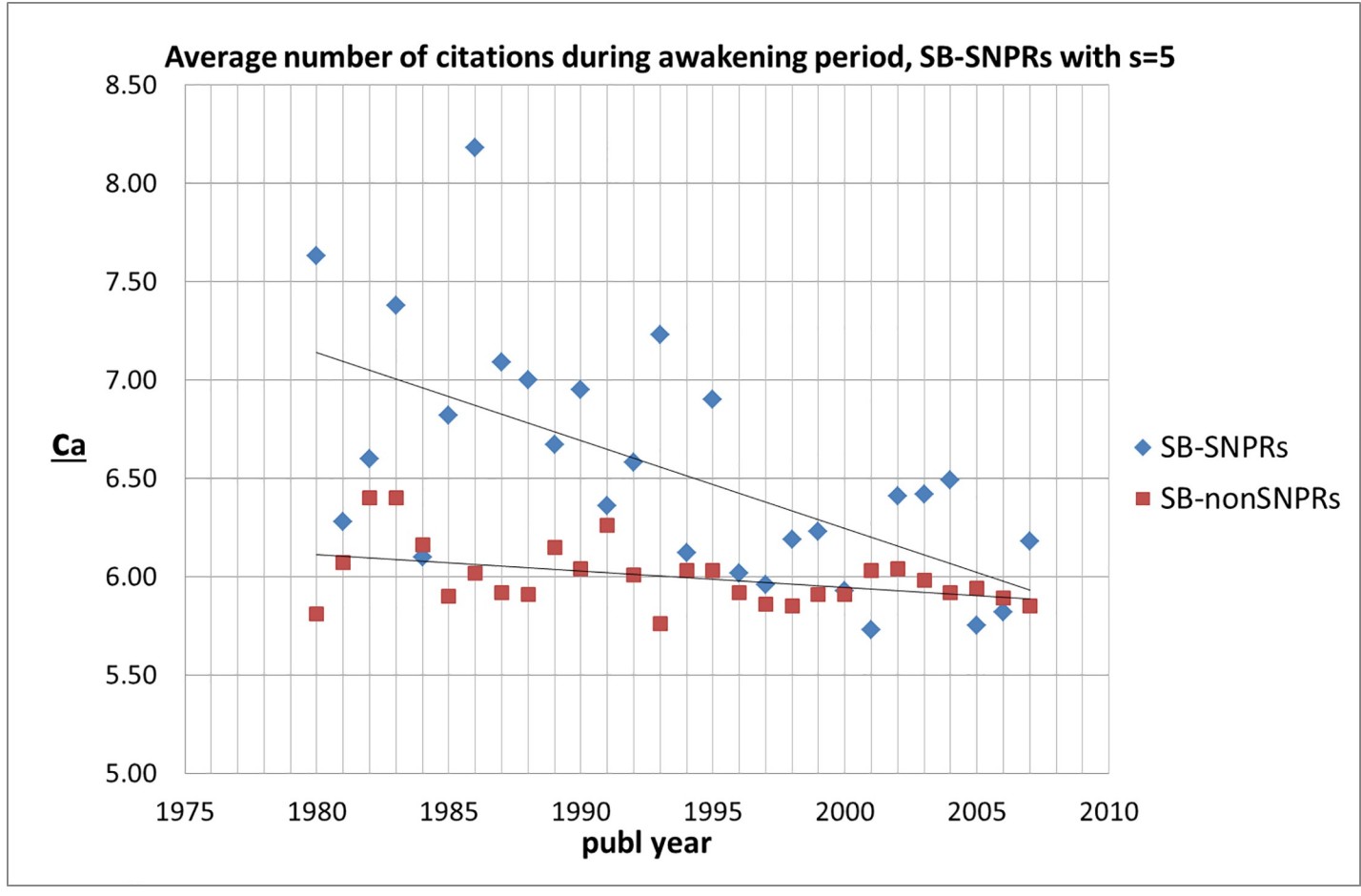

**Fig 22. Average awakening citation-intensity $c_a$ for SN-SNPRs and SB-nonSNPRs with $s = 5$.**

see Figs 23 and 24 which show the sleeping period followed by an exponential increase of citations. Fig 24 also compares the citation trend of the Matthews SB-SNPR [24] with an average SB-SNPR published in 1985. The citation rate of this SB-SNPR during sleeping time was $c_s = 1.0$ ('deep sleep') and the citation rate during the awakening period was $c_a = 7.0$ which is just slightly above the average of all 5,261 SBs with $s = 5$. But at the end of the awakening period an exponential increase of citations starts. From 2008 until now the paper has more than 1,000 citations per year, with a total of 18,141 (WoS Core Collection, August 14, 2018) which makes this SB-SNPR currently the third highest cited paper of all WoS-covered papers published in 1985 and places this SB-SNPR within the top-50 highest cited papers ever.

This SB-SNPR of Matthews and co-authors published in the journal Diabetologia deals with a mathematical, computerized model to determine plasma glucose and insulin concentrations by their interaction in a feedback loop. This model enables to accurately predict the homeostatic concentrations which arise from varying degrees of cell function deficiency and insulin resistance. The five main research fields of the citing papers are endocrinology & metabolism (38%), nutrition dietetics (13%), general internal medicine (6%), cardiac & cardiovascular systems (6%), peripheral vascular diseases (6%). The journal Diabetologia ranks 16 by journal impact in the field Endocrinology & Metabolism. So we see that a very highly cited paper is not necessarily a paper in a top-journal in terms of, say, the top-5 in journal impact.

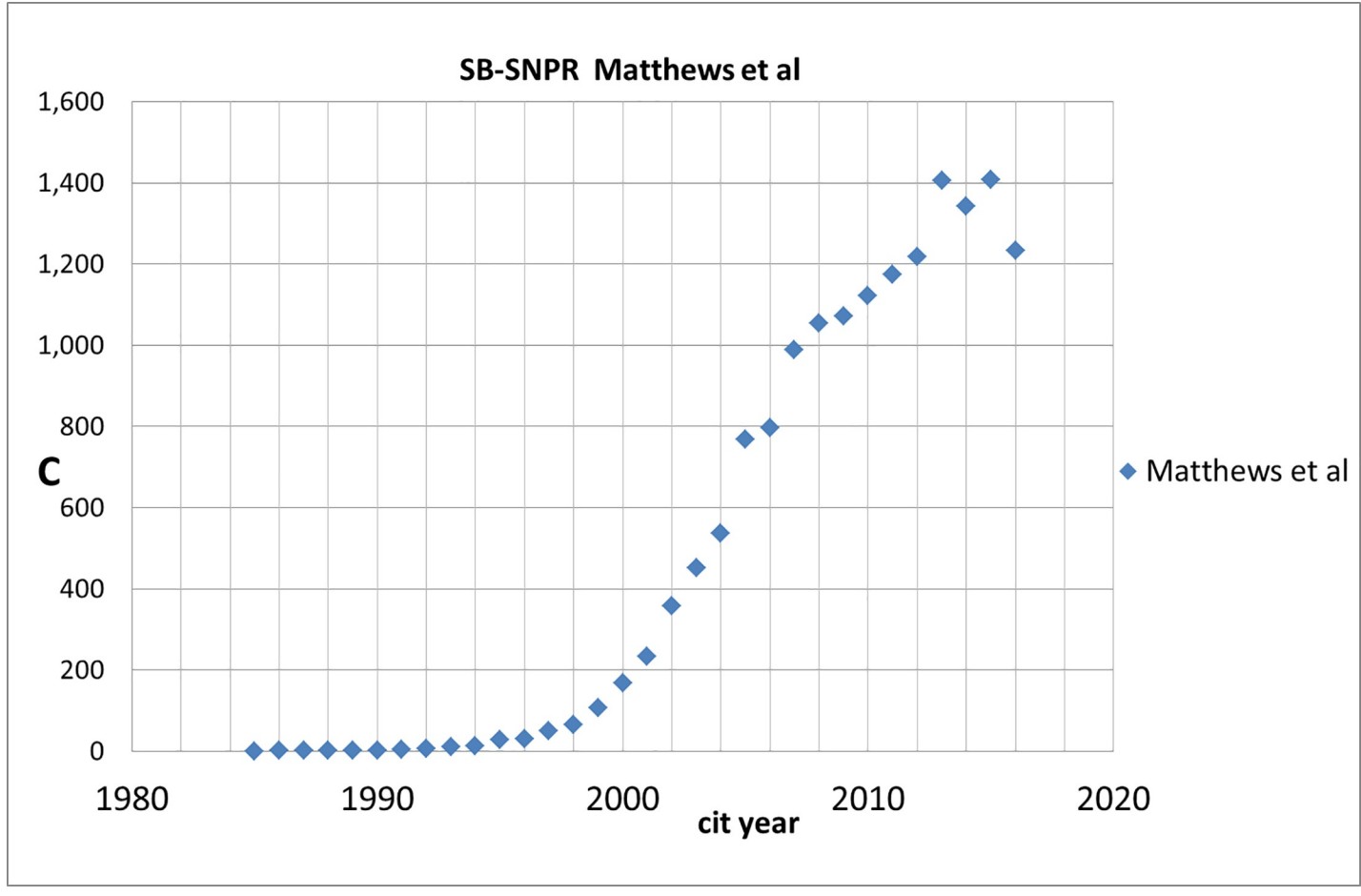

**Fig 23. Trend in the number of citations received by the Matthews et al [24], an SB-SNPR published in 1985.** On the abscissa the years of citation are given, on the ordinate the number of citations.

From the 18,141 papers that cite the Matthews SB-SNPR we selected the 500 most cited of which we made a co-citation analysis and mapping with the CWTS VoSviewer [25]. For a detailed discussion of the use of the mapping tool VoSviewer in our Sleeping Beauties research we refer to our earlier paper [6].

The results are shown in Fig 25. The co-citation clusters of the references of the top-500 citing papers are from a variety of medical fields which is nicely reflected by the clusters in the butterfly-like structures with different colors. The green cluster represents research on the metabolic syndrome and cardiovascular diseases; the purple cluster relates to insulin resistance, obesity and the role of hormones; the blue cluster also focuses on insulin resistance but particularly liver diseases; the red cluster focuses on predicting and assessing insulin resistance; and the yellow cluster relates to genetic studies of diabetes. Undoubtedly the Matthews SB-SNPR is the most central paper as it is, by definition, cited by all citing papers.

The VoSviewer also enables us to measure all co-occurrences of any possible pair of concepts (author- and/or database-given keywords, or terms parsed from titles and abstracts) in a set of papers. In this way we created a map in which the conceptual structure of the research represented by the set of the top-500 papers citing the Matthew SB-SNPR. The results are shown in Fig 26. We clearly see concept clusters representing different themes. Major clusters are the blue one around glucose, insulin resistance, and obesity; the green cluster around

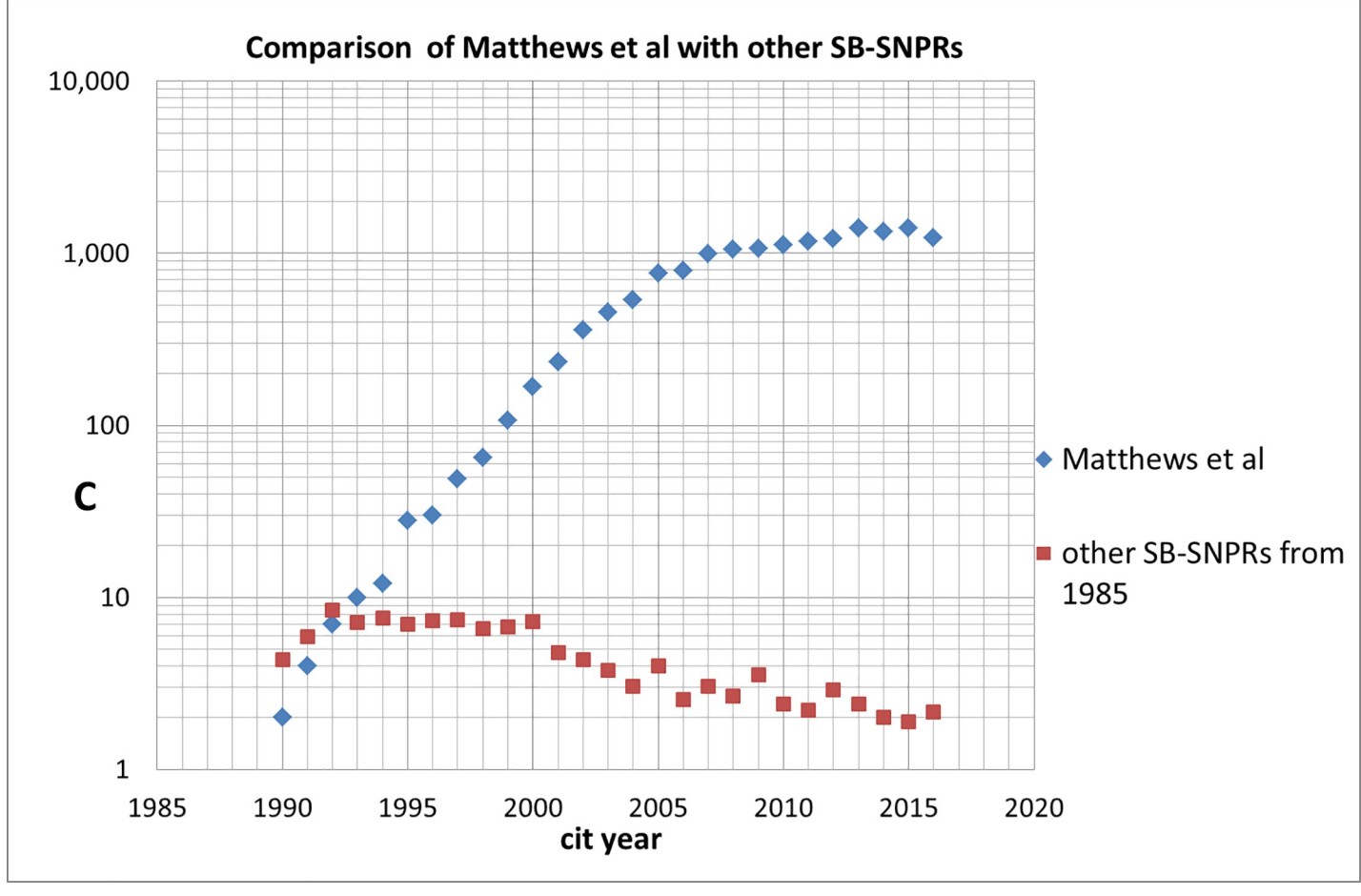

**Fig 24. Trend in the number of citations received by the Matthews et al [24] compared to the average of the other SB-SNPRs published in 1985.** On the abscissa the years of citation are given, on the ordinate the number of citations.

diabetes, beta-cell function and homeostasis; the red cluster around cardiovascular diseases. Clearly insulin resistance plays a central role.

Even within 10 years after the publication of the Matthews SB-SNPR there are no citing patents. The first patent citation is in 2001, 16 years after the publication of this SB-SNPR. As we have seen in Fig 23, the number of (scientific) citations has already started to rise. It takes until 2010 before the number of citing patents increases rapidly. In the PATSTAT database (Spring version 2016) 39 citing patents are registered, several of these patents are highly cited by other patents. The patents are from the following technology fields: pharmaceuticals, organic fine chemistry, basic materials chemistry, biotechnology, analysis of biological materials, medical technology. Notice the difference between the citing scientific and the citing technological fields. Perhaps surprisingly, the patents fields most clearly show where the Matthews SB-SNPR is about, particularly analysis of biological materials; the scientific fields show where the work of Matthews et al is applied.

## Conclusions

In this paper we investigate in the medical research fields recent Sleeping Beauties with a special focus on those SBs that are cited in patents (SB-SNPRs). We find that the increasing trend

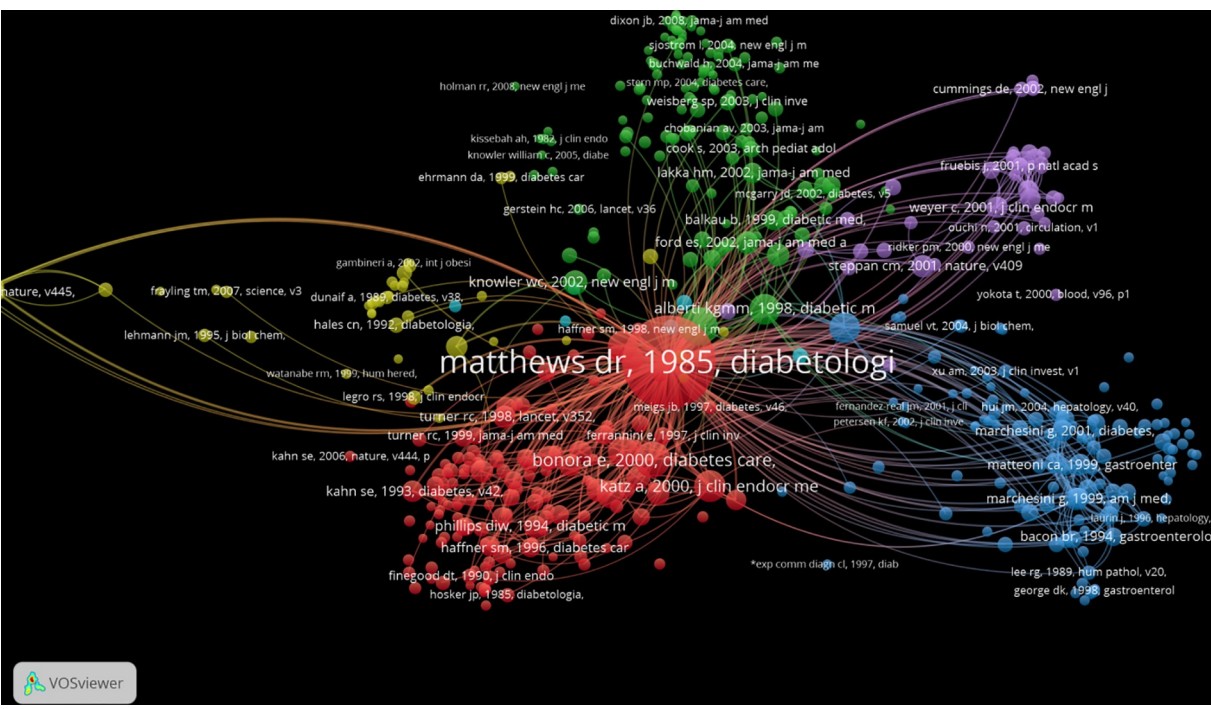

**Fig 25. Co-citation map of the 500 most highly cited papers that cite the Matthews et al SB-SNPR (co-citation threshold = 3).**

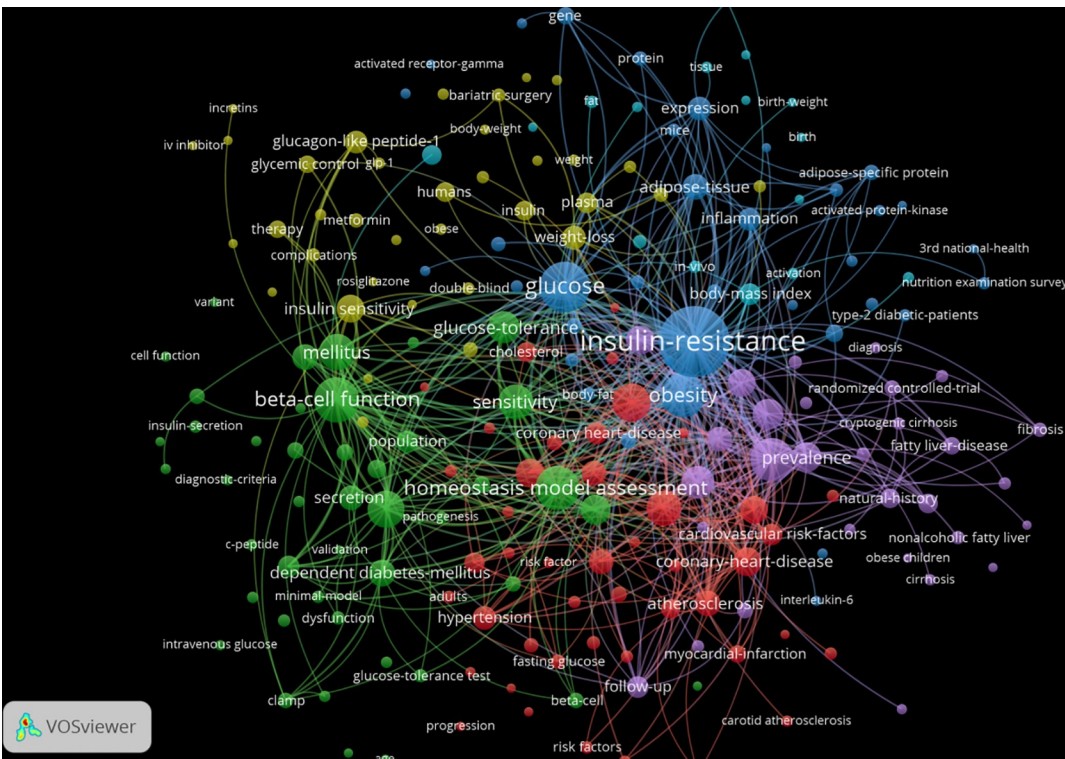

**Fig 26. Concept co-occurrence map of the 500 most highly cited papers that cite the Matthews et al SB-SNPR (co-occurrence threshold = 5).**

of the relative number of SBs comes to an end around 1998. It confirms our earlier observations in the natural science and engineering fields and supports our conjecture that the expanding worldwide facilities to access scientific publications seems to have stopped increasing trends in the occurrence of SBs. Apparently, however, this does not prevent that a more or less constant fraction of publications still becomes an SB.

We measured for the entire period 1980–2007 the scaling of the number of SBs with sleeping period length, with during-sleep citation-intensity and with the awakening citation-intensity. Particularly the scaling with sleeping period length shows a remarkable time-dependent change: during our measuring period the scaling exponent doubled. Our explanation of this change is that in the early 1980s the probability to have SBs with very long sleeping times was considerably higher than it is nowadays. On the basis of these scaling measurements we determined the *Grand Sleeping Beauty Equation*. This equation describes important quantitative characteristics of SBs. If a publication is *twice as longer in deep sleep*, the probability that it awakes with, on average, at least five citation per year during five years, is about an *order of magnitude less*. In other words, the longer the sleeping period, the less probable it is that a publication will awake. Indeed, a finding that we can intuitively recognize. Next, the probability to find a publication that has twice as much citations in deep sleep (e.g., 4 versus 2 in five years sleep), is about factor 2.5 higher. In other words, the less deep the sleep, the more SBs will be found, because we are moving toward 'normal' publications. The probability for a higher awakening citation-intensity decreases very rapidly. For instance, the probability that a SB will have during the awakening period a citation intensity twice as large as another SB (e.g., 10 versus 5 citations per year in the five years awakening period), is about a factor 50 lower.

Similar to citations given by publications, also the number of citations given to SBs by patents is characterized by a skewed distribution. Our study demonstrates that for SB-SNPRs with $s$ = 5 the fraction for at least one patent citation within 5 years after publication, and thus within their sleeping period, is in the 1980s about 25%. But about 20 years later, in the first decade of this century, the situation has changed drastically: 70–80% of SB-SNPRs are cited within their sleeping period. For SB-SNPRs with the longer sleep period $s$ = 10 the fraction for at least one patent citation within 5 years after publication is in the 1980s about 10% and in the first decade of this century it is 40%, again a substantial change. It turns out that the fraction of SB-SNPRs that are cited by patents *within their sleeping period* is exponentially increasing and after the year 2000 practically all SB-SNPRS are cited by patents within their sleeping period.

To investigate the increasingly faster pace of technological impact in more detail, we analyzed the time lag between the filing year of the patent that is the *first* citer of the SB-SNPR and the publication year of the SB-SNPR. This *technological time lag* ranges for the SB-SNPRs from 1 to 30 years. We find that for SB-SNPRs with $s$ = 5 the technological time lag remains rather stable in the 1980s but after 1990 it becomes rapidly shorter (with about 7% per year) and around 2002 it becomes shorter than the sleeping time. The technological time lag becomes already in the early 1990s shorter than the sleeping time for the SB-SNPRs with $s$ = 10. This can be expected given the similarity of the trend of the technological time lag for both the SB-SNPRs with $s$ = 5 as well as with $s$ = 10. Like our earlier findings for the natural sciences and engineering these observations again suggest that, on average, in the more recent years the majority of SB-SNPRs are cited by one or more patents before the 'scientific wakening'.

A new element in this study is the time trend of inventor-author self-citations. With inventor-author self-citations we mean that at least one of the *inventors of patents citing* SB-SNPRs is also an *author of the cited SB-SNPR*. This number increases (SB-SNPRs with $s$ = 5) with about 11% per year: in recent years the occurrence of inventor-author self-citations is about four to five times higher than in the 1980s. We find that inventor-author self-citations may result in shorter technological time lags, but this effect is small. The findings in this study on

inventor-author self-citations in medical SB-SNPRs confirm the results of our earlier studies in the natural sciences and engineering that inventor-author self-citation is quite rare. In these earlier studies we found on the basis of a number of individual cases that the probability of earlier *scientific* awakening triggered by inventor-author self-citation is small. In this study we find that this is the case for the medical research fields too.

We also studied the technological impact of the patents citing SB-SNPRs by analyzing the extent to which these patents themselves are cited later on by other patents within ten years after the filing date of the patent. This enabled us to identify the top-20% highly cited patents. Remarkably, the fraction of SB-SNPRs with citing patents belonging to the top-20% remains more or less constant over the entire measuring period.

For both the SB-SNPRs as well as the SB-nonSNPRs with $s$ = 5 we found no significant correlation between the number of citations by other publications during the sleeping period and the number of citations by other publications during the awakening period. In other words, the depth of the sleep is no predictor for the awakening intensity. Remarkably, however, we do find that the average number of citations (by other publications) during the awakening period tends to be higher for the SB-SNPRs ($s$ = 5), but this difference disappears gradually. We also did not find a significant correlation between the awakening intensity and the technological impact of patents that cite the SB-SNPR. This means that the scientific impact of Sleeping Beauties is generally not related to the technological importance of the SBs, as far as measured with number and impact of the citing patents. Again this is similar to our earlier findings for the natural sciences and engineering.

Within our set of medical Sleeping Beauties one SB stands out, the Matthews SB-SNPR, published in 1985, on a mathematical, computerized model to determine plasma glucose and insulin concentrations, a method which turned out to be of great significance for diabetes patients. Even within 10 years after the publication of this SB-SNPR there are no citing patents. The first patent citation is in 2001, 16 years after the publication of this SB-SNPR. It takes until 2010 before the number of citing patents and thus the *technological impact* of this SB-SNPR increases rapidly. In the PATSTAT database Spring version 2016 39 citing patents are registered, several of these patents are highly cited by other patents. But most striking is the delayed but enormously increasing *scientific* impact of this SB-SNPR: it took more than five years before the Matthews SB-SNPR started to become reasonably cited (more than 5 citations per year), then the number of citations increased exponentially and from 2008 until now the paper has more than 1,000 citations per year, with a total of nearly 20,000 which makes this SB-SNPR currently the third highest cited paper of all WoS-covered papers published in 1985 and places this SB-SNPR within the top-50 highest cited papers ever.

## Supporting information

**S1 Fig. Average citation trend in the period 2000–2016 for SBs with $s$ = 5 and 10 (both $c_s$(max) = 1.0) as well an average medical publication with publication year 2000.**
(TIF)

**S2 Fig. Scaling exponent β of during-sleep citation intensity (SB-SNPRs with $s$ = 10), each five-years block is located in the figure by its middle year.**
(TIF)

**S3 Fig. Scaling exponent γ of awakening citation-intensity (SB-SNPRs with $s$ = 10), each five-years block is located in the figure by its middle year.**
(TIF)

**S1 Table. WoS Fields codes and names of medical research fields.**
(DOCX)

**S2 Table. Numbers of the identified SBs.**
(DOCX)

**S3 Table. Trend of real and normalized number of SBs with $s = 5$ ($c_s$(max) = 1) in the medical research fields, numbers are based on successive five-years blocks.**
(DOCX)

**S4 Table. Number of all publications from $s = 1$ to $s = 20$, with $c_s$(max) = 1, $c_a$(min) = 5, $a$(min) = $a$(max) = 5, within period 1980–2008.**
(DOCX)

**S5 Table. Number of SBs ($s = 5$) for successive during-sleep citation-intensity intervals.**
(DOCX)

**S6 Table. Number of SBs ($s = 10$) for successive during-sleep citation-intensity intervals.**
(DOCX)

**S7 Table. Number of SBs ($s = 5$) for awakening citation-intensity intervals.**
(DOCX)

**S8 Table. Number of SBs ($s = 10$) for awakening citation-intensity intervals.**
(DOCX)

## Author Contributions

**Conceptualization:** Anthony F. J. van Raan.

**Data curation:** Anthony F. J. van Raan, Jos J. Winnink.

**Formal analysis:** Anthony F. J. van Raan.

**Investigation:** Anthony F. J. van Raan.

**Methodology:** Anthony F. J. van Raan, Jos J. Winnink.

**Resources:** Jos J. Winnink.

**Software:** Jos J. Winnink.

**Supervision:** Anthony F. J. van Raan.

**Writing – original draft:** Anthony F. J. van Raan.

**Writing – review & editing:** Jos J. Winnink.

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
