## [Decision Letter · Decision Letter 0]

9 Aug 2019

PONE-D-19-15090

The Occurrence of ‘Sleeping Beauty’ Publications in Medical Research: Their Scientific Impact and Technological Relevance

PLOS ONE

Dear Professor van Raan,

Thank you for submitting your manuscript to PLOS ONE. After careful consideration, we feel that it has merit but does not fully meet PLOS ONE’s publication criteria as it currently stands. Therefore, we invite you to submit a revised version of the manuscript that addresses the points raised during the review process. We would appreciate receiving your revised manuscript by Sep 23 2019 11:59PM. To enhance the reproducibility of your results, we recommend that if applicable you deposit your laboratory protocols in protocols.io, where a protocol can be assigned its own identifier (DOI) such that it can be cited independently in the future. For instructions see: http://journals.plos.org/plosone/s/submission-guidelines#loc-laboratory-protocols

We look forward to receiving your revised manuscript.

Kind regards,

Lutz Bornmann

Academic Editor

PLOS ONE

Journal Requirements:

Reviewers' comments:

Reviewer's Responses to Questions

**Comments to the Author**

1. Is the manuscript technically sound, and do the data support the conclusions?

Reviewer #1: Yes

Reviewer #2: Yes

2. Has the statistical analysis been performed appropriately and rigorously? 

Reviewer #1: Yes

Reviewer #2: Yes

3. Have the authors made all data underlying the findings in their manuscript fully available?

Reviewer #1: No

Reviewer #2: Yes

4. Is the manuscript presented in an intelligible fashion and written in standard English?

Reviewer #1: Yes

Reviewer #2: Yes

5. Review Comments to the Author

Reviewer #1: The manuscript "The Occurrence of 'Sleeping Beauty' Publications in Medical Research: Their Scientific Impact and Technological Relevance" presents an analysis of sleeping beauties (papers with delayed recognition) in medical research fields and a subsequent analysis of citations from patents to the sleeping beauties. Herewith the authors continue their previous work on this topic. In general, the manuscript is well written and interesting. There are a few often occurring inconsistencies, e.g., awake vs. awakening, SB vs. SBs, Sleeping Beauty vs. SB, and "Fig " vs. "Fig. ". Many commas after introductory phrases are missing, e.g., "In follow-up work [7, 8] we ..." should read "In follow-up work [7, 8], we ...".

I see a problem with the open data policy of PLoS One and the data used in this study. I disagree with the authors' statement regarding data availability. I suggest that the authors share a small test data set (maybe from CrossRef) along with their "efficient search algorithm written in SQL" so that readers can reproduce the work in principle.

The authors should add a very recent study on a closely related subject to the introduction:

https://arxiv.org/abs/1906.07953

The authors choose to use "a five-years awake period". As mentioned above, this probably should be "a five-years awakening period". Furthermore, it would be good to provide a reason for this arbitrary choice.

The definition of "ca" and "cs" becomes problematic with the equations, e.g., does "A.cs" mean A times cs or A times c times s? From the context, it becomes clear that the former is the case, but it would be clearer if ca and cs would have "a" and "s" as subscript.

How were "medical research fields" defined for this study, by a list of WoS subject categories or research areas?

The "normalization on the basis of the growth factors given in Table 2" (p. 5) should be explained in more detail. Is it simply N(s)/(growth factor)?

There is a strange sentence on page 7: "In this way, we are able to determine the probability that an SB occurs with specific values of the three above variables occurs." Maybe, it's fixed when the second "occurs" is removed.

Page 10: "In our 2004-study we ... ." Better provide the reference.

Also page 10: "Again this has to done for each sleeping period separately ... with a different sleeping periods ... ."  "Again this has to be done for each sleeping period separately ... with different sleeping periods ... ."

Page 14 and 20/21: "... than de sleeping time ..."  "... than the sleeping time ..."

Also page 14: "We intend to focus on these development ...". Either "We intend to focus on this development ..." or "We intend to focus on these developments ..."

Page 16: "These 59 SB-SNPRs are cited by 206 patents, of which nearly half, 95, deal with inventor-author self-citation." This sounds like the 95 patents have inventor-author self-citations as a topic. Please rephrase.

Page 18: "... ten tear ..."  "... ten years ..."

Caption of Fig. 22: "... SN=SNPRs ..."  "... SB-SNPRs ..."

Page 19: Please define more explicitly what do mean with concepts.

In the case of Fig. 25 and Fig. 26, webstartable links could be provided so that the interested reader could zoom into the maps.

Page 21 and also in similar occasions, it would be better to refer to "the Matthews SB-SNPR" with a formal reference or "Matthews et al.".

Unclear sentence on page 21: "It takes until 2010 before the number of citing patents and with that the technological impact of this SB-SNPR increases rapidly, in the PatStat database Spring version 2017 39 citing patents are registered, several of these patents are highly cited by other patents." Please reword, probably it is best to split the sentence.

Axis labels of all figures should be revised. Axis labels, such as "publ y", "s", "cit y", "C", and "pub y" are not helpful to readers.

Reviewer #2: 1. Is the manuscript technically sound, and do the data support the conclusions? YES

2. Has the statistical analysis been performed appropriately and rigorously? YES

The first author had in 2004 introduced the concept of “Sleeping Beauties in Science” - scientific papers receiving only delayed recognition. i.e. average citation numbers below certain thresholds during specified time periods after publication (periods of more or less “deep sleep”) and average citation numbers above certain thresholds during specified time periods after the sleep period. The distribution of SBs had been studied in its (power-law) dependence on three parameters: the length of the sleep in years, the depth of the sleep and the intensity of the awakening (discovery), both measured by averaged annual citation numbers. The author had thereby arrived at a “Grand Sleeping Beauty Equation”.

For more recent publications the authors had applied this methodology to SBs especially in physics, chemistry, and engineering & computer science and moreover on scientific SBs cited in patents.

The current study is in a sense a sequel of these recent publications in that it focuses on the field of medical research and therein on SBs cited in patents. This is clearly stated by the authors in their introduction, where they explain the main research questions and key claims and provide the context especially by referring to own previous work.

This study – as the authors rightly conclude – confirms many of the findings of their previous works, but is also able to resolve differences of the scientific areas under investigation.

For the sake of comparability to their previous studies the authors very decidedly and successfully followed very similar procedures for data collection and data analysis. But they also go beyond and additionally investigate time trends in the data. e. g. concerning inventor- author self-citations, or - with a further indirection - the citations to patents which cite scientific SBs.

-> Minor issue: Fig 3 does not force the interpretation given in the first paragraph of page 6. For the case s=5 the graph would also allow for ongoing oscillations with a net increase.

3. Have the authors made all data underlying the findings in their manuscript fully available? YES

But with the following qualifications:

The authors refer to an in-house Web of Science derived custom database, which in this form is not available to the public. But other research institutes may have comparable access to the basic data, which presuppose a subscription - and should in principle be able to reproduce the results - which are given in more numerical detail in a supplement - or even apply the methodology with another research focus.

-> Minor issue: To that end it would be helpful to disclose the “fast and efficient search algorithm written in SQL”.

At the end of the results section the authors zoom in on the outstanding SB in their data set and its unique properties in the set of scientific SBs cited in patents. To that end they apply analyses of co-citations and co-occurrences of keywords, using the VOSviewer.

-> Minor issue: In order to view the resp. graphs properly the authors should provide the corresponding network files.

4. Is the manuscript presented in an intelligible fashion and written in standard English? YES

-> Minor issues: But it is recommended to go through the whole article and eliminate some distracting typos, grammatical and syntactical errors and incorrect references to table or figure numbers and numerical values given in figures.

6. PLOS authors have the option to publish the peer review history of their article (what does this mean?). If published, this will include your full peer review and any attached files.

Reviewer #1: No

Reviewer #2: No

---

## [Author Response · Author response to Decision Letter 0]

28 Aug 2019

Dear Editor,

This writing is our revised cover letter in response to your comments and the comments of the reviewers. We first give a short summary of the subject of research, its importance, the applied methods, and the results (this is in main lines the same as in our first cover letter). Next we address the points you made; for our response to the comments of the reviewers and the amendments we made on the basis of these comments we refer to the ‘Reponses to the reviewers’. 

This manuscript presents the results of our study on scientific papers in medical research that go unnoticed for a long time and then almost suddenly attract attention ('Sleeping Beauties in Science', SBs). In order to compare characteristics of SBs in the medical fields with those in the natural sciences and engineering (discussed in our earlier related papers), we apply exactly the same measuring protocol and data analysis. In our research on medial SBs we found new phenomena. We here mention three striking ones. The first is found by an extensive statistical analyses of the scaling of the number of SBs with sleeping period length, during-sleep citation-intensity, and with awake citation-intensity. With these scaling data we determined the Grand Sleeping Beauty Equation (GSBE). This GBSE confirms our earlier measurement but the new finding is that the scaling exponents also show a time-dependent behavior which suggests a decreasing occurrence of SBs with longer sleeping periods. A second novel finding is that by far the largest part of the medical SBs becomes very highly cited publications, they even belong to the top-10 to 20% most cited publications in medicine. Also new is our observation that in recent years the occurrence of inventor-author self-citations is about four to five times higher than in the 1980’s. We conclude the paper with a remarkable example of a medical SB that became, and still is to this day, a very highly cited paper. The manuscript presents an investigation of scientific work in medical research fields that is often ahead of its time and receives a delayed recognition. We think such a theme is interesting for a broader PLOS audience. 

Regarding your remarks: we carefully studied the reviewers’ comments and consider these comments to be particularly valuable. They really helped to improve the paper. On the basis of these comments we revised the text and the figures where appropriate. As said above, our responses to the reviewers’ comments are in the file ‘Reponses to the reviewers’. In addition, the changes in the papers are indicated in the file ‘Revised Manuscript with Track Changes’. 

Data availability: relevant data will be available from the Open Science Framework (osf.io/4ru96). 

The authors

Leiden, August 27, 2019.

---

## [Decision Letter · Decision Letter 1]

20 Sep 2019

The Occurrence of ‘Sleeping Beauty’ Publications in Medical Research: Their Scientific Impact and Technological Relevance

PONE-D-19-15090R1

Dear Dr. van Raan,

We are pleased to inform you that your manuscript has been judged scientifically suitable for publication and will be formally accepted for publication once it complies with all outstanding technical requirements.

With kind regards,

Lutz Bornmann

Academic Editor

PLOS ONE

Additional Editor Comments (optional):

Reviewers' comments:

Reviewer's Responses to Questions

**Comments to the Author**

1. If the authors have adequately addressed your comments raised in a previous round of review and you feel that this manuscript is now acceptable for publication, you may indicate that here to bypass the “Comments to the Author” section, enter your conflict of interest statement in the “Confidential to Editor” section, and submit your "Accept" recommendation.

Reviewer #1: All comments have been addressed

Reviewer #2: All comments have been addressed

2. Is the manuscript technically sound, and do the data support the conclusions?

Reviewer #1: Yes

Reviewer #2: Yes

3. Has the statistical analysis been performed appropriately and rigorously? 

Reviewer #1: Yes

Reviewer #2: Yes

4. Have the authors made all data underlying the findings in their manuscript fully available?

Reviewer #1: No

Reviewer #2: Yes

5. Is the manuscript presented in an intelligible fashion and written in standard English?

Reviewer #1: Yes

Reviewer #2: Yes

6. Review Comments to the Author

Reviewer #1: (No Response)

Reviewer #2: (No Response)

7. PLOS authors have the option to publish the peer review history of their article (what does this mean?). If published, this will include your full peer review and any attached files.

Reviewer #1: No

Reviewer #2: No

---

## [Editor Report · Acceptance letter]

9 Oct 2019

PONE-D-19-15090R1 

The Occurrence of ‘Sleeping Beauty’ Publications in Medical Research: Their Scientific Impact and Technological Relevance 

Dear Dr. van Raan:

I am pleased to inform you that your manuscript has been deemed suitable for publication in PLOS ONE. Congratulations! Your manuscript is now with our production department. 

With kind regards,

on behalf of

Dr. Lutz Bornmann 

Academic Editor

PLOS ONE